# Exposure Assessment of Cadmium in Female Farmers in Cadmium-Polluted Areas in Northern Japan

**DOI:** 10.3390/toxics8020044

**Published:** 2020-06-17

**Authors:** Hyogo Horiguchi, Etsuko Oguma, Satoshi Sasaki, Kayoko Miyamoto, Yoko Hosoi, Akira Ono, Fujio Kayama

**Affiliations:** 1Department of Hygiene, Kitasato University School of Medicine, Kanagawa 252-0374, Japan; oguma@med.kitasato-u.ac.jp (E.O.); a-ono@furukawadenchi.co.jp (A.O.); 2Department of Environmental and Preventive Medicine, School of Medicine, Jichi Medical University, Tochigi 329-0498, Japan; yk_hosoi2005@yahoo.co.jp (Y.H.); kayamafujio@gmail.com (F.K.); 3Department of Social and Preventive Epidemiology, School of Public Health, The University of Tokyo, Tokyo 113-0033, Japan; stssasak@m.u-tokyo.ac.jp; 4Department of Registered Dietitian, Koyo Nursing Nutrition College, Koyo Gakuen, Ibaraki 306-0013, Japan; kayokomy@nifty.com; 5Environmental Promotion Department, The Furukawa Battery Co., Ltd., Fukushima 972-8501, Japan

**Keywords:** cadmium, food, farmer, PTWI (provisional tolerable monthly intake), TWI (tolerable weekly intake), Monte Carlo simulation

## Abstract

Akita prefecture is located in the northern part of Japan and has many cadmium-polluted areas. We herein performed an exposure assessment of cadmium in 712 and 432 female farmers in two adjacent cadmium-polluted areas (A and B, respectively), who underwent local health examinations from 2001–2004. We measured cadmium concentrations in 100 food items collected from local markets in 2003. We then multiplied the intake of each food item by its cadmium concentration in each subject to assess cadmium intake from food and summed cadmium intake from all food items to obtain the total cadmium intake. Median cadmium intake levels in areas A and B were 55.7 and 47.8 µg/day, respectively, which were both higher than that of the general population and were attributed to local agricultural products, particularly rice. We also calculated weekly cadmium intake per body weight and compared it to the previous provisional tolerable weekly intake reported by the Joint FAO (Food and Agriculture Organization)/WHO (World Health Organization) expert committee on food additives or current tolerable weekly intake in Japan of 7 µg/kg BW/week. Medians in areas A and B were 7.2 and 6.0 µg/kg BW/week, respectively. Similar estimated values were also obtained by the Monte Carlo simulation. These results demonstrated that the cadmium exposure levels among the farmers were high enough to be approximately the tolerable weekly intake.

## 1. Introduction

Humans are exposed on a daily basis to cadmium (Cd), a toxic heavy metal, mainly through the consumption of food containing Cd. Cd absorbed via food accumulates in the kidneys and may cause renal tubular dysfunction, called Cd nephropathy, in the inhabitants of Cd-polluted areas [1]. The most severe case of Cd toxicity is itai-itai disease, which is characterized by osteomalacia and renal anemia. It develops among patients with Cd nephropathy [2]. The heaviest Cd-polluted area in Japan was along the Jinzu River basin of Toyama prefecture, at which agricultural fields were contaminated by a large amount of Cd derived from an upstream mine. A large number of inhabitants in this area developed Cd nephropathy and 200 patients with itai-itai disease were officially recognized by 2020. 

Although the Cd-polluted area in Toyama was completely restored in 2011, large but scattered Cd-polluted areas still remain in Akita prefecture, which is located in the northern part of Japan, due to the previous activities of mines and smelters [3]. Pollution levels are particularly high in the northern area of Akita prefecture, in which we performed local health examinations on female farmers as part of the Japanese multi-centered environmental toxicant study (JMETS) [4,5]. JMETS aimed for risk assessment of Cd by targeting females, who are generally more vulnerable to Cd toxicity than males. Health examinations were sequentially performed in two adjacent areas: the area along Yoneshiro River in Odate city in 2001–2002 (area A) [4] and that upstream of the river in Kazuno city and Kosaka town in 2003–2004 (area B) [5] (Figure 1). Cd pollution levels were higher in area B than in area A because area B was directly affected by two large mines and their affiliated smelters, while area A was secondarily contaminated through irrigation from the river that runs from area B. These studies revealed that many farmers were exposed to high levels of Cd through the consumption of self-harvested rice contaminated by Cd, some of whom had Cd nephropathy [4,5,6].

The provisional tolerable monthly intake (PTMI) of Cd reported by the Joint FAO (Food and Agriculture Organization)/WHO (World Health Organization) expert committee on food additives (JECFA) is now 25 µg/kg/month and was amended from the provisional tolerable weekly intake (PTWI) of 7 µg/kg/week in 2010 [7]. In Japan, the tolerable weekly intake (TWI) of Cd of 7 µg/kg/week remains in effect and was set by the Food Safety Commission of Japan (FSCJ) in 2008. In our previous study on area A in 2001, we performed dietary exposure assessments on Cd in subjects undergoing health examinations and obtained weekly Cd intake levels for comparisons with the PTWI of Cd at that time [4]. The findings revealed that 33–51% of subjects had Cd intake levels in excess of the PTWI. However, the exposure assessment was performed using a simplified method based on Cd concentrations in rice and miso, fermented soybean paste, and the consumption of these two foods to estimate individual Cd intake according to average Japanese Cd intake levels. Therefore, a more detailed exposure assessment of Cd is needed in this area to clarify the actual status of dietary Cd exposure in its inhabitants. A similar exposure assessment of Cd in area B, in which Cd pollution was heavier than in area A, needs to be conducted. 

In the present study, we collected 100 types of local food items in 2 Cd-polluted areas in Akita, measured their Cd concentrations, and assessed Cd exposure levels in the subjects of previous local health examinations using an individual food analysis method based on Cd concentrations in individual food items and the intake amounts of these food items by subjects obtained from diet surveys conducted at health examinations. We also performed the Monte Carlo simulation [8] to evaluate the probabilistic distribution of the Cd intake levels of these subjects in order to confirm the assessment. We demonstrated that Cd intake levels in these subjects were approximately PTWI or TWI, which showed that subjects exposed to Cd were at risk of adverse effects. 

## 2. Materials and Methods 

### 2.1. Sampling and Handling of Food Items

We selected approximately 80 local food items from the table of food groups of the Japan’s National Health and Nutrition Survey provided by the Ministry of Health, Labor, and Welfare, that were eaten at a high frequency in diet surveys from health examinations conducted on local female farmers in area A in 2001–2002 [4]. We then deleted food items in which Cd concentrations were undetectable in the study on the absorption rate of dietary Cd among female farmers conducted in the winter of 2002–2003 [9]. We added food items that were assumed to have high Cd concentrations, such as seaweed, shellfish, mollusks, and livers, based on a previous study [10]. In November and December 2003, we purchased these food items at local markets in areas A and B based on the recommendations of a local female farmer who was familiar with traditional dietary patterns. We collected 100 food items for the measurement of Cd concentrations. We aimed to collect three of each food item in each area; however, this number increased or decreased depending on the inventory status. Therefore, the number of purchases ranged between 1 and 10, with an average of six. We collected 100 g of the edible portions of each food item and stored them at room temperature, 4 °C, or −20 °C depending on their perishability until the measurement of Cd concentrations. 

### 2.2. Measurement of Cd Concentrations

The measurement of Cd concentrations in food items, except for rice, was conducted by Japan Food Research Laboratories (Tokyo, Japan). Next, 10 to 20 g of the edible portion of food, which was precisely measured in a Kjeldahl flask, was added to 200 mL of nitric acid (HNO_3_) and heated. After the vigorous reaction was completed, 5 mL of sulfuric acid (H_2_SO_4_) was added to the flask and heated again until the color changed to light yellow. After cooling, the inside of the flask was washed well with deionized water (less than 1 µs/cm of conductance) and heated again until H_2_SO_4_ was released as white smoke. The residual was then dissolved in an appropriate quantity of deionized water to make a sample solution. The solution was moved to a separatory funnel and 10 mL of 50% diammonium hydrogen citrate and Thymol blue indicator (0.1 g of Thymol blue in 100 mL of ethyl alcohol) were then added. After neutralization with ammonium solution, the volume of the solution was increased to 100 mL by the addition of deionized water, and this was followed by 5 mL of 3% ammonium pyrrolidine-N-dithiocarbamate (APDC) solution/ammonium sulfate and 10 mL of butyl acetate with shaking for 5 min. After being left to stand, the butyl acetate layer was collected for the measurement of Cd concentrations using a flame atomic absorption spectrometer (AA-890, Nippon Jarrell-Ash Co., Ltd., Tokyo, Japan). The original standard Cd solution (Kanto Chemical Co., Inc., Tokyo, Japan) was diluted with 1% hydrochloric acid (HCl) to make 0.4 and 0.8 µg/mL standard Cd solutions. Quality control was achieved in the analysis using sugar (commercial products) added with Cd as an alternative to certified reference materials. Its additional recovery was maintained within 90–110%. The detection limits for cereals, other food items, and drinking water were 0.01 mg/kg, 0.005 mg/kg, and 0.001 mg/L, respectively. 

### 2.3. Health Examinations and Diet Surveys

Health examinations performed on female farmers in areas A and B in 2001–2004 were described previously [4.5]. In the present study, we used data obtained on age, height, weight, and the results of a diet history questionnaire (DHQ) from 725 and 438 subjects in areas A and B, respectively. DHQ is designed to assess food and nutrient intake levels in the previous month based on the quantity and semiquantitative frequency of the consumption of 110 food items commonly eaten in Japan [11]. Estimates of intake for food, energy, and selected nutrients were calculated using an ad hoc computer algorithm for DHQ based on Standard Tables of Food Composition in Japan, which has already been validated [12,13]. We used data obtained on individual food intake levels to calculate Cd intake levels. Among subjects, 12 and 6 with extremely low or high energy intake levels (≤1000 or ≥3500 kcal/day) were excluded in areas A and B, respectively, in addition to one whose consumption of rice was zero in area A, resulting in 712 and 432 subjects for analyses. 

### 2.4. Calculation of Cd Intake Levels

We calculated the Cd intake levels of individual subjects by multiplying the intake of each food item by its Cd concentration and then summed Cd intake levels from all foods consumed. However, the food items for which intake levels were assessed by DHQ and those for which Cd concentrations were measured were not always in one-to-one correspondence. Therefore, we adjusted mismatches for reconciliation, as described below. Regarding boiled barley-rice, which is assumed to consist of 70% rice and 30% wheat, the Cd concentrations of rice and wheat flour, respectively, were multiplied. Since the intake levels of udon (wheat noodles) and soba (buckwheat noodles) were collectively assessed by DHQ, they were divided into two halves, each of which was multiplied by the Cd concentration of udon or soba. The intake levels of various noodles, including Chinese noodles or spaghetti, which were separately assessed by DHQ, were multiplied by the Cd concentration of udon, assuming that these foods were similarly made of wheat flour. The intake levels of butter rolls, croissants, pizza, pancakes, and okonomiyaki (Japanese-style pancakes) were also multiplied by the Cd concentration of white bread, while those of snack foods, Japanese sweets, cakes, cookies, and doughnuts were multiplied by the Cd concentration of manju (Japanese sweet bun). Regarding sweet potatoes, taro, yams, and Chinese yams, which were assessed collectively by DHQ, the average Cd concentration of sweet potatoes, taro, and yams was used. The average Cd concentration of silken and cotton tofu was used for tofu (soybean curd). The average Cd concentration of spinach, garland chrysanthemum, Japanese mustard spinach, Bok choy (Chinese cabbage), and Japanese leek was used for leafy green vegetables, which were collectively assessed by DHQ. Concerning the intake of wakame seaweed, the Cd concentration of raw wakame seaweed was used. The intake of mushrooms was evenly divided into shiitake mushroom and other mushrooms, and the Cd concentrations of shiitake mushroom and maitake mushroom were multiplied. Regarding the intake of shrimp and fish eggs, the average Cd concentrations of prawns and shrimp and of cod and salmon roe, respectively, were used. Since the intake of shellfish included oysters and other shellfish in DHQ, the Cd concentrations of oysters with innards and the average Cd concentrations of scallops without innards, Japanese littleneck clams, and freshwater clams were respectively multiplied. The average Cd concentration of beef liver, pork liver, chicken liver, Hinai chicken liver, and the innards of Hinai chicken was used for the intake of liver. Food items for which Cd concentrations were not detected, such as animal meat, eggs, milk, and green tea (after brewing), were not included in the calculation of total Cd intake. Cd intake from garlic, okra, belvedere fruit, kelp, hijiki seaweed, agar-agar, mozuku seaweed, and scallops with innards was also excluded because they were not evaluated by DHQ; however, their Cd concentrations were measured. The arithmetic means (AMs) of the Cd concentrations of these food items from both areas were multiplied by their corresponding food intake, while rice Cd concentrations in individual subjects, which were obtained in previous health examinations, were used to calculate individual Cd intake from rice. When a subject consumed brown rice, the Cd concentration of which is generally reduced by 10% due to polishing [14], the Cd concentration adjusted to be equivalent to polished rice was multiplied for calculations. One rice sample was missing in area A, which was substituted with the geometric mean (GM) of the Cd concentration in rice. The Cd concentrations of some foods with masses that are changed by cooking were corrected using the table of mass changes in individual food items from the Standard Tables of Food Compositions in Japan (7th edition, Japanese Ministry of Education, Culture, Sports, Science and Technology).

### 2.5. Statistical Analysis

Age, height, weight, total energy intake, and rice intake, which followed a normal distribution, were presented as AMs with arithmetic standard deviations (SDs) and differences in their mean values were analyzed by the Student’s *t*-test. Since Cd concentrations in rice followed a clear lognormal distribution with large numbers and the distribution of Cd concentrations in other food items was not clear because of small numbers, rice and other food items were shown as GMs and AMs, respectively. Cd intake levels with skewed distributions were presented as medians with 25th and 75th percentiles and differences in their median values were analyzed by the median test. The *χ^2^* test was used to compare the proportions of Cd intake per body weight above the PTWI or TWI. Regarding multiple comparisons, the Steel-Dwass test was performed to compare the medians of age-classified Cd intake. The judgment of outliers was made using the Smirnov-Grubbs test. After the exclusion of outliers, the Monte Carlo simulation was performed on weekly Cd intake levels with 10,000 repetitions of the calculation based on the supposition of a lognormal distribution. Statistical analyses were performed using IBM SPSS Statistics V25 (SPSS Japan, Tokyo, Japan) based on the basic management of data by Mac Excel Tokei ver. 2.0 (Esumi, Tokyo, Japan).

## 3. Results

Food items and their Cd concentrations, divided into 10 subgroups, are listed in Table 1, Table 2, Table 3, Table 4, Table 5, Table 6, Table 7, Table 8, Table 9 and Table 10. Cd concentrations in rice and rice products were high (Table 1). The average Cd concentrations in rice, 0.158 and 0.109 mg/kg, were lower than the safety standard of 0.4 mg/kg; however, 8.2 and 5.8% of rice had Cd concentrations that were above the safety standard in areas A and B, respectively. Among cereals, tubers, and roots, while the Cd concentrations of wheat flour and its products were not high, taro (satoimo in Japanese) had a high Cd concentration of 0.289 mg/kg (Table 2). The Cd concentrations of soybeans, including edamame, were high, whereas those of their processed foods were not, except for miso (Table 3). Cd was detected in all vegetables investigated, among which spinach, Japanese parsley, garland chrysanthemum, Japanese mustard spinach, and belvedere fruit showed high Cd concentrations (Table 4). Shiitake mushroom and seaweed (wakame, kombu, nori, and hijiki in Japanese) had markedly high Cd concentrations (Table 5 and Table 6). Among fish and shellfish, salted squid guts, scallops with innards, oysters with innards, and freshwater clams, all of which had innards, had very high Cd concentrations, while fish meat itself did not (Table 7). Cd was not detected in many livestock food items, such as meat, eggs, and milk, except for the innards (Table 8). Cd was not detected in fruit (Table 9). High Cd concentrations were found in chocolate and tea leaves (Table 10). Cd was not detected in brewed tea.

We then calculated Cd intake levels by subjects who underwent health examinations, multiplied Cd concentrations in food items by individual food intake, and showed the results obtained in subgroups. The backgrounds of subjects in areas A and B, the food intake levels for whom were used to calculate Cd intake levels, are shown in Table 11. No significant differences were observed in age, energy intake, or rice intake between the 2 areas, whereas significant differences were noted in height (*p* = 0.047) and weight (*p* = 0.047), but were biologically negligible. 

Cd intake from seaweed, fish, and shellfish was combined into one subgroup as seafood and that from livestock food was included in the subgroup as others, while that from fruit was excluded from calculations (Table 12). Among the subgroups, Cd intake from the subgroup of rice and rice products was the highest in both areas, and accounted for approximately 40–50% of the total Cd intake. Cd intake levels from the subgroups of vegetables and seafood were higher than those in the other subgroups. Cd intake from the subgroup of rice and rice products was significantly higher in area A than in area B, while those from other subgroups were similar between the two areas, except for vegetables, with Cd intake being significantly lower in area A than in area B. The median total Cd intake levels in areas A and B were 55.7 and 47.8 µg/day, respectively, with the former being significantly higher than the latter.

We then compared the results obtained with Cd intake by the general population in Japan (Figure 2) [15,16]. Cd intake has been gradually decreasing in Japan: 31.1 µg/day (16.2 µg/day from rice and 14.9 µg/day from other food items) in 1981, 21.1 µg/day (7.8 µg/day from rice and 13.3 µg/day from other food items) in 2007, and 17.8 µg/day (5.7 µg/day from rice and 12.1 µg/day from other food items) in 2015. This was mainly attributed to a reduction in Cd intake from rice. Cd intake levels from rice and rice products in areas A (28.3 µg/day) and B (19.4 µg/day) were 3.6- and 2.5-fold higher, respectively, than that (7.8 µg/day) by the general population in 2007, while Cd intake levels from other food items in areas A and B (approximately 24.0 µg/day) were 1.8-fold higher than that (13.3 µg/day) by the general population. Total Cd intake levels in areas A and B were approximately 2.5-fold higher than that by the general population.

We also calculated weekly Cd intake per body weight, using individual values of body weight, to compare the PTWI of JECFA at that time or the current TWI in Japan of 7 µg/kg BW/week. Median weekly Cd intake levels were 7.2 and 6.0 µg/kg BW/week in areas A and B, respectively, with both being approximately PTWI or TWI (Table 13). The distributions of weekly Cd intake levels in areas A and B were shown in histograms (Figure 3). The exclusion of data obtained from two subjects in area A that were extremely large and considered to be outliers resulted in similar distributions in both areas that skewed to the higher side. The percentages of subjects with weekly Cd intake levels above PTWI or TWI were 51.7 and 38.0% in areas A and B, respectively (*p* < 0.05, *χ^2^* test) (Table 13). 

We further divided weekly Cd intake per body weight into age-classified groups and examined differences between them (Table 14). In area A, weekly Cd intake per body weight was higher in subjects aged 40 or older than in younger subjects, while no significant differences were observed between age-classified groups in area B. Weekly Cd intake per body weight in subjects aged 50 or older was above PTWI or TWI in area A. 

We then performed the Monte Carlo simulation using the same data and the results obtained are shown in Figure 4. We excluded 2 outliers in area A, which markedly skewed their probability density distributions. As a result, we obtained estimated median weekly Cd intake levels of 7.0 and 6.0 µg/kg BW/week in areas A and B, respectively (Table 15), which were similar to the results described above. 

## 4. Discussion

In the present study, we collected food items from local markets in two Cd-polluted areas in Akita, Japan, measured their Cd concentrations, and attempted to accurately assess Cd intake levels in local female farmers who had been exposed to high Cd levels by calculating individual Cd intake based on data obtained from diet surveys performed in health examinations. The present results demonstrated that Cd intake was approximately 2.5-fold higher in our subjects than in the general population. Furthermore, weekly Cd intake per body weight, which was confirmed by the Monte Carlo simulation, were approximately the same as the PTWI of JECFA or TWI of Japan, while 38.0–51.7% of subjects had weekly Cd intake per body weight that was above it. 

The high Cd intake levels observed among our subjects may be attributed to high Cd concentrations in local agricultural products, particularly rice. Although Cd concentrations in rice in these areas (GMs of 0.158 and 0.109 mg/kg in areas A and B, respectively) were lower than the safety standard (0.4 mg/kg), they were markedly higher than the median rice Cd concentrations in Japan (0.06 mg/kg in 1997 and 1998 and 0.05 mg/kg in 2009 and 2010) [16], and there were rice that had Cd concentrations above the safety standard (8.2 and 5.8% in areas A and B, respectively). Although the intake of Cd was higher in area A than in area B, the actual accumulated levels of Cd in subjects were higher in area B than in area A, as demonstrated by their blood and urinary Cd levels (for example, urinary Cd levels in 70–79-year-old subjects were 4.90 and 9.34 µg/g cr. in areas A and B, respectively, and 2.99 µg/g cr. in control subjects) [5]. Since the biological half-life of Cd in humans is very long (10–30 years), blood and urinary Cd levels are stable indicators of the accumulation of Cd in the human body, particularly when the level of Cd that accumulates is high [1]. This discrepancy may have been due to the timing of the initiation of measures in rice farming to lower Cd absorption from the soil in 2002, namely, the flooding of paddy fields before and after heading during August [5,17]. Actually, the rates of rice with Cd concentration above the safety standard in these areas were decreased after that according to the results of intensive inspections targeted for Cd-polluted rice fields by Akita prefecture (Akita Prefect. Department of of Agriculture, Forestry, and Fisheries) (Appendix A). More recently, our own investigations on farmers in the areas show that the median of Cd concentration in their self-harvested rice from 2010 to 2018 was 0.096 mg/kg and 0.7% of them were above the safety standard (*n* = 599) (unpublished data). These results indicate the flooding of paddy fields have effectively lowered Cd concentrations in rice from 2002 in these areas. The health examination in area A was performed in 2001–2002, at the initiation of the flooding of paddy fields, while that in area B was performed in 2002–2003, just after the flooding of paddy fields had started. Therefore, Cd intake levels may have been higher in area B than in area A before the start of the flooding of paddy fields. 

Cd intake levels from food items other than rice and rice products, which remained constant independent of the flooding of rice paddy fields, were also higher in these areas in Akita than the average in Japan (Figure 2) [15]. Although total Cd intake other than rice and rice products in these areas was nearly twice the average in Japan in 2007; those from other food items varied [15]. Cd intake from cereals, tubers, and roots was 2.3–2.4 µg/day in these areas, which is similar to the average in Japan of 2.7 µg/day. On the other hand, Cd intake from soybeans and soybean products was 3.4 µg/day in these areas, which was approximately 3-fold higher than the average in Japan of 1.1 µg/day. Cd intake from vegetables and seafood in the general Japanese population was reported in the classification as “brightly colored vegetables”, “vegetables, seaweeds”, and “seafood” by the Ministry of Agriculture, Forestry, and Fisheries. Therefore, we compared total Cd intake from vegetables and seafood in areas A and B with total Cd intake from “brightly colored vegetables”, “vegetables, seaweeds”, and “seafood” in the general Japanese population; the former, 12.3–13.1 µg/day, was higher than the latter, 8.2 µg/day. In addition, Cd intake from mushrooms in these areas, 2.6–2.8 µg/day, which was absent from the report by the Ministry of Agriculture, Forestry, and Fisheries of Japan, significantly contributed to total Cd intake. These comparisons revealed that agricultural products in these areas contained high Cd concentrations due to Cd contamination in the local farmland, and the consumption of these products was a significant contributor to Cd overexposure among local farmers. 

Among agricultural products, Cd concentrations were high in sesame seeds, taro (satoimo in Japanese), yams (yamaimo in Japanese), soybeans, carrot, spinach, burdock (gobo in Japanese), garland chrysanthemum (shungiku in Japanese), Japanese mustard spinach (komatsuna in Japanese), garlic, belvedere fruit (tomburi in Japanese), and shiitake mushroom. Among seafood, many types of seaweed (such as wakame, kombu, and hijiki in Japanese), salted squid guts, scallops (hotate in Japanese) and oysters (kaki in Japanese) with innards, Japanese littleneck clam (asari in Japanese), and freshwater clam (shijimi in Japanese) showed very high Cd concentrations. On the other hand, Cd concentrations were generally low in livestock food and fruit. These results are consistent with previous findings showing that Cd concentrations were high in tubers, soybeans, brightly colored vegetables, and seafood, particularly seaweed, and the innards of squid or shellfish in Japan [10,15].

Median weekly Cd intake levels in our subjects were 7.2 and 6.0 µg/kg BW/week in areas A and B, respectively, which were approximately the PTWI of JECFA at that time or the current TWI in Japan of 7 µg/kg BW/week. In our previous study on area A, we estimated weekly Cd intake levels to be between 5.70–6.72 µg/kg BW/week, which was consistent with the present results, based on 2 assumptions: all food items other than rice may be contaminated by Cd at the same percentage contribution as rice (50%) or at a constant level (15.0 µg/day) [4]. In addition, the percentage of subjects with weekly Cd intake levels higher than 7 µg/kg BW/week in area A, 51.7%, in the present study was similar to previously estimated percentages, 33–51%. These results indicate that previously estimated Cd intake levels, based only on Cd concentrations in rice and miso, were not inaccurate, and that local farmers in Cd-polluted areas in Akita exposed to Cd were at risk of adverse effects. Weekly Cd intake per body weight was above the PTWI or TWI in older subjects in area A, suggesting a higher risk of developing renal tubular dysfunction among the elderly. There were no age-classified subgroups above the PTWI or TWI in area B, but actually the older subjects might had been exposed to much higher levels of Cd like in area A before the start of the flooding of paddy fields.

Two methods are generally employed in exposure assessments of chemicals in food: a total diet study (TDS) and individual food analysis. There are two approaches in TDS: a market basket method and duplicate portion study [5,18]. In the market basket method, which assesses chemical intake in divided food groups, the average intake of a chemical in a certain population may be calculated; however, the concentration of this chemical in individual food items remains unknown. On the other hand, the duplicate portion study, which measures chemical concentrations in whole meals, provides information on actual chemical intake by individuals, but does not give stable results on chemical intake. We simultaneously performed a duplicate portion study in area A on 17 female farmers for three days to support the present results, and obtained a smaller median Cd intake of 19.0 µg/day that ranged between 9.8–63.1 µg/day (unpublished data). In contrast, the individual food analysis, adopted in the present study, allowed us to identify the source foods of chemical intake and calculate the average intake of a chemical in a population, similar to the market basket method. The individual food analysis generally cannot exclude uncertainties due to processing and cooking foodstuffs. However, Cd itself does not increase or diminish in foodstuffs by processing and cooking. Furthermore, mass changes in individual foods were included in the calculation of Cd intake based on the tables of mass changes in individual food, from the Standard Tables of Food Compositions in Japan (7th edition, Japanese Ministry of Education, Culture, Sports, Science, and Technology). Furthermore, a probabilistic assessment, the Monte Carlo simulation, may be performed using individual Cd concentrations in food items. 

The National Institute of Health Sciences previously investigated Cd intake by the general Japanese population using the market basket method [15] and an assessment of Cd intake based on the individual food analysis, similar to the present study, was not previously performed in Japan. Cd intake in the Jinzu River basin of Toyama, the heaviest Cd-polluted area in Japan, was reported to be as high as 600 µg/day in 1968 using the market basket method [19]. On the other hand, in Kosaka town in Akita, the intake of Cd from rice by local residents, assessed from Cd concentrations in rice, was 100.6 µg/day in 1974–1976 [20], 92 µg/day in 1978, and 55 µg/day in 1999–2000 using the duplicate portion study [19]; the latter was similar to the present results. These findings indicate that Cd intake in Cd-polluted areas in Akita has been decreasing for decades, and are consistent with the present results. Cd intake in Cd-polluted areas in Akita is still higher than those by the general Japanese population and Western counties. Recent Cd intake by general populations assessed using the market basket method was 4.63 μg/day or 0.54 μg/kg body weight/week in U.S.A. [21], 0.16 μg/kg body weight/week in France [22], 0.77 μg/kg body weight/week in Spain [23], 1 μg/kg body weight/week in Sweden [24], 5.00 μg/day in Italy [25], 0.85μg/kg body weight/week in Belgium [26], and 13.5 μg/day in Denmark [27]. Although Cd intake in Eastern Asian countries, where rice is consumed as a staple food, is generally high, such as 32.7 μg/day in China [28] and 22.0 μg/day in Korea [29], Cd intake levels remain higher in Akita. 

There are some limitations in the present study. The data used in this study were obtained in 2001–2003. Nevertheless, the results obtained remain important, even after approximately 20 years, because few Cd-polluted areas remain in Japan. Furthermore, data were collected at the start of the flooding of paddy fields, which has successfully decreased Cd exposure levels in farmers in these areas; therefore, these data will never be obtained again in the future. 

Furthermore, the subjects examined were females and therefore the Cd intake by male farmers remains unknown. Cd intake levels by males may be higher than those by females based on differences in the amount of food consumed. However, since females are generally more vulnerable to Cd toxicity than males, such as higher intestinal Cd absorbability in females than in males, it is not inappropriate to use the results of the present study for risk assessments of Cd in the general population. In area B, the results of the Cd exposure assessment did not reflect previous Cd intake because of the flooding of paddy fields. However, the Cd intake level in area B was still higher than that by the general population. In DHQ, which assesses general nutritional intake, some food items that showed markedly higher Cd concentrations were absent, such as kelp and hijiki. Therefore, we were unable to include Cd intake from these foods in total Cd intake, which may have led to the underestimation of Cd exposure. The Japanese government provides Cd intake values by the general population as averages. Although median values in Cd-polluted areas cannot be statistically compared to these averages, the differences are large enough to identify areas at risk of health problems. 

## 5. Conclusions

We performed an exposure assessment of Cd in female farmers, who are more vulnerable to Cd toxicity than males in Cd-polluted areas in Akita, Japan. Participants underwent local health examinations during 2001–2004, using the individual food analysis method with the Monte Carlo simulation. Results showed that Cd intake was higher than that by the general population, which was derived from local agricultural products, particularly rice, and also that their exposure levels to Cd were approximately the PTWI of JECFA or TWI of Japan. 

## Figures and Tables

**Figure 1 toxics-08-00044-f001:**
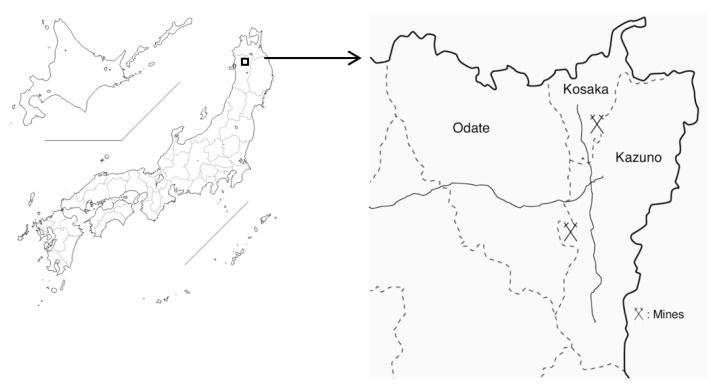
A map of area A (Odate city) and area B (Kazuno city and Kosaka town) in Akita, and their location in Japan (cited from CraftMAP). The thick solid line, dotted lines, and thin solid lines indicate the prefectural boundary, boundaries of municipalities, and courses of the river, respectively.

**Figure 2 toxics-08-00044-f002:**
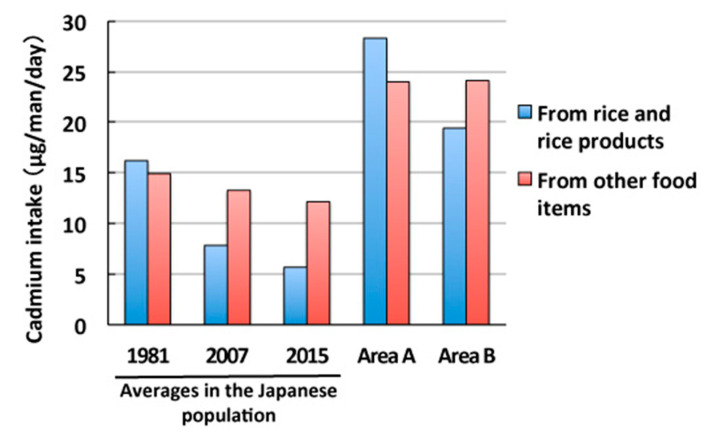
Cadmium intake levels per person from rice and rice products and from others in female farmers in cadmium-polluted areas A and B in Akita, Japan, shown by medians, and comparisons with average cadmium intake in the Japanese population.

**Figure 3 toxics-08-00044-f003:**
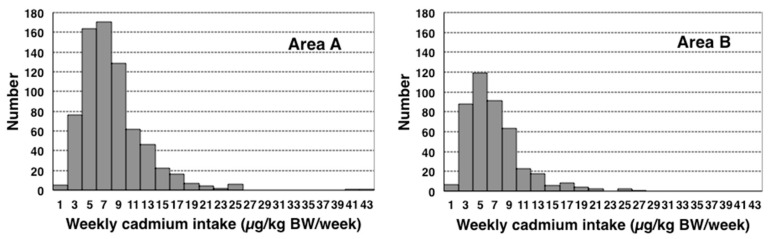
Distribution of weekly cadmium intake per body weight in female farmers in cadmium-polluted areas A and B in Akita, Japan.

**Figure 4 toxics-08-00044-f004:**
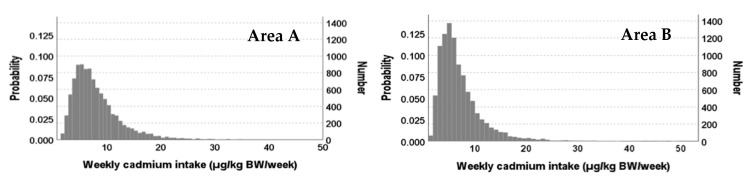
Probability density distributions of weekly cadmium intake per body weight in female farmers in cadmium-polluted areas A and B in Akita, Japan, estimated using the Monte Carlo simulation.

**Table 1 toxics-08-00044-t001:** Cadmium (Cd) concentrations in rice and rice products. Data are presented as arithmetic means, except for rice, which are presented as geometric means (GMs).

Food Items	*n*	Cd (mg/kg)	Ranges
Rice (area A)	711 *	0.158 (GM)	<0.02–0.971
Rice (area B)	432	0.109 (GM)	0.008–0.687
Kiritampo; pounded rice skewer	6	0.063	0.070–0.102
Glutinous rice	10	0.098	0.02–0.32
Rice cakes	5	0.069	0.017–0.182
Rice crackers	4	0.091	0.017–0.263

*: One sample is missing.

**Table 2 toxics-08-00044-t002:** Cadmium (Cd) concentrations in cereals, tubers, and roots. Data are presented as arithmetic means.

Food Items	*n*	Cd (mg/kg)	Ranges
White bread	6	0.018	0.016–0.023
Ampan, bean-jam bun	6	0.010	0.010–0.014
Wheat flour *	3	0.021	0.019–0.024
Udon; wheat noodles	6	0.005	0.005–0.007
Soba; buckwheat noodles	6	0.019	0.007–0.034
Sesame seeds	6	0.055	0.021–0.11
Sweet potato	6	0.008	0.005–0.016
Potato	6	0.034	0.005–0.098
Taro	6	0.289	0.036–0.795
Yams, Chinese yams	7	0.061	0.005–0.167
Potato chips *	3	0.059	0.029–0.114

*: Only from area A.

**Table 3 toxics-08-00044-t003:** Cadmium (Cd) concentrations in soybeans and soybean products. Data are presented as arithmetic means.

Food Items	*n*	Cd (mg/kg)	Ranges
Soybeans	4	0.115	0.05–0.25
Silken tofu	5	0.015	0.009–0.028
Cotton tofu	6	0.027	0.013–0.052
Deep-fried tofu	6	0.051	0.033–0.083
Natto; fermented soybeans	6	0.028	0.011–0.055
Miso; soybean paste	6	0.123	0.026–0.259
Edamame; soybeans in the pod	6	0.155	0.085–0.293
Soy sauce	4	0.019	0.015–0.026

**Table 4 toxics-08-00044-t004:** Cadmium (Cd) concentrations in vegetables. Data are presented as arithmetic means.

Food Items	*n*	Cd (mg/kg)	Ranges
Carrot	6	0.047	0.013–0.107
Spinach	6	0.064	0.030–0.122
Tomato	5	0.012	0.005–0.022
Squash	6	0.016	0.008–0.024
Broccoli	6	0.012	0.005–0.027
Japanese white radish	6	0.009	0.005–0.023
Onions	5	0.016	0.005–0.031
Cabbage	6	0.008	0.006–0.011
Chinese cabbage	6	0.020	0.011–0.038
Burdock	6	0.063	0.017–0.212
Dropwort Japanese parsley	6	0.010	0.005–0.019
Eggplant	6	0.017	0.005–0.028
Garland chrysanthemum	5	0.074	0.006–0.244
Japanese mustard spinach	5	0.065	0.009–0.23
Bok choy (Chinese cabbage)	5	0.029	0.01–0.101
Green pepper	5	0.006	0.005–0.009
Garlic	6	0.051	0.01–0.142
Okra	3	0.023	0.012–0.043
Belvedere fruit	6	0.069	0.041–0.095
Japanese leek *	3	0.032	0.005–0.083
Pickled vegetables	10	0.022	0.009–0.095
Smoked daikon pickles	6	0.024	0.017–0.035

*: Only from area A.

**Table 5 toxics-08-00044-t005:** Cadmium (Cd) concentrations in mushrooms. Data are presented as arithmetic means.

Food Items	*n*	Cd (mg/kg)	Ranges
Raw shiitake mushroom	7	0.374	0.065–0.527
Maitake mushroom	6	0.043	0.023–0.108

**Table 6 toxics-08-00044-t006:** Cadmium (Cd) concentrations in seaweed and seaweed products. Data are presented as arithmetic means.

Food Items	*n*	Cd (mg/kg)	Ranges
Wakame seaweed (raw)	6	0.253	0.069–0.544
Wakame seaweed (dried) *	3	4.64	4.11–5.06
Kelp (konbu seaweed)	6	0.682	0.119–1.78
Laver (nori seaweed)	4	0.413	0.209–0.66
Hijiki seaweed	5	1.066	0.693–1.53
Agar-agar *	3	0.024	0.014–0.033
Mozuku seaweed	6	0.006	0.005–0.01

*: Only from area A.

**Table 7 toxics-08-00044-t007:** Cadmium (Cd) concentrations in fish and shellfish. Data are presented as arithmetic means.

Food Items	*n*	Cd (mg/kg)	Ranges
Salmon	6	<0.005	
Tuna	6	0.009	0.005–0.013
Cod	6	<0.005	
Horse mackerel	6	0.012	0.007–0.015
Mackerel	6	0.012	0.005–0.018
Sandfish without eggs	6	0.012	0.009–0.017
Sandfish with eggs	6	0.014	0.01–0.018
Squid	6	0.032	0.018–0.081
Salted squid guts	5	2.36	0.978–6.57
Octopus	6	0.007	0.005–0.011
Prawn	7	0.055	0.005–0.17
Shrimp*	2	0.022	0.009–0.035
Cod roe	6	0.008	0.005–0.015
Salmon roe	3	<0.005	
Scallops without innards	5	0.0408	0.012–0.103
Scallops with innards	5	3.635	0.684–5.54
Oysters with innards	6	0.680	0.486–1.03
Japanese littleneck clam	4	0.160	0.028–0.305
Freshwater clam	4	0.375	0.235–0.55
Hampen (fish minced and steamed)	6	0.005	0.005–0.006
Dried whitebait	6	0.010	0.005–0.02
Broiled eel	6	0.008	0.005–0.011

*: Only from area B.

**Table 8 toxics-08-00044-t008:** Cadmium (Cd) concentrations in livestock food. Data are presented as arithmetic means.

Food Items	*n*	Cd (mg/kg)	Ranges
Beef	6	<0.005	
Pork	5	<0.005	
Chicken	4	<0.005	
Hinai chicken	5	<0.005	
Horse meat	3	0.006	0.005–0.007
Beef liver *	1	0.021	
Pork liver	4	0.024	0.016–0.028
Chicken liver	5	0.015	0.01–0.022
Hinai chicken liver *	2	0.039	0.033–0.044
Innards of Hinai chicken	6	0.021	0.008–0.066
Sausage	3	0.006	0.005–0.008
Egg	6	<0.005	
Milk	6	<0.005	

*: Only from area A.

**Table 9 toxics-08-00044-t009:** Cadmium (Cd) concentrations in fruit. Data are presented as arithmetic means.

Food Items	*n*	Cd (mg/kg)	Ranges
Apple	6	<0.005	
Apple juice	6	<0.005	
Kiwi fruit	5	<0.005	

**Table 10 toxics-08-00044-t010:** Cadmium (Cd) concentrations in others. Data are presented as arithmetic means.

Food Items	*n*	Cd (mg/kg)	Ranges
Manju; sweet bun	10	0.022	0.010–0.127
Chocolate *	3	0.042	0.03–0.084
Curry roux *	3	0.015	0.012–0.018
Flavor seasonings *	1	0.025	
Ketchup	3	0.016	0.016–0.017
Japanese green tea leaves *	3	0.026	0.009–0.049
Nutritional supplement drink *	3	<0.005	
Well water *	7	<0.005	

*: Only from area A.

**Table 11 toxics-08-00044-t011:** Backgrounds of female farmers who underwent health examinations in areas A and B.

Backgrounds	Area A (*n* = 712)		Area B (*n* = 432)	
	Mean ± SD	Ranges	Mean ± SD	Ranges
Age	57.4 ± 11.3	21–79	57.2 ± 9.3	35–82
Height (cm)	152.0 ± 6.2	130–180	152.8 ± 5.9 *	132–169
Weight (kg)	54.5 ± 8.0	34–92	55.5 ± 8.5 *	33–94
Energy intake (kcal/day)	1933.7 ± 463.7	1000–3440	1923.8 ± 459.0	1047–3451
Rice intake (g/day)	371.8 ± 118.9	78.6–1120	359.0 ± 105.3	30–880

*: *p* < 0.05 versus area A (unpaired Student’s *t*-test).

**Table 12 toxics-08-00044-t012:** Daily cadmium intake per person (µg/day) in female farmers in areas A and B.

Subgroups of Cadmium Intake	Area A (*n* = 712)		Area B (*n* = 432)	
	Median (25–75th Percentile)	Ranges	Median (25–75th Percentile)	Ranges
Total cadmium intake	55.7 (40.5–75.4)	10.6–301	47.8 (34.2–64.5) *	10.2–187
Rice and rice products	28.3 (17.1–44.7)	0.1–289	19.4 (9.9–37.4) *	0.1–154
Cereals, tubers, and roots	2.4 (1.5–4.3)	0–40.7	2.3 (1.4–3.8)	0–13.8
Soybeans and soybean products	3.4 (2.3–4.6)	0.2–22.6	3.4 (2.5–4.5)	0.3–11.1
Vegetables	6.2 (3.9–9.0)	0.5–26.2	6.8 (4.3–9.4) *	0.7–29
Mushrooms	2.8 (1.1–4.4)	0–39.3	2.6 (0.8–4.4)	0–24.6
Seafood **	6.1 (3.4–10.9)	0.3–46.5	6.3 (3.9–9.9)	0.4–52.2
Others ***	0.5 (;0.3–0.9)	0–6.1	0.5 (0.3–0.9)	0–5.4

*: *p* < 0.05 versus area A (median test). **: including seaweed, fish, and shellfish. ***: including manju, livestock food, chocolate, and flavor seasonings.

**Table 13 toxics-08-00044-t013:** Weekly cadmium intake per body weight (µg/kg BW/week) in female farmers in areas A and B and their distribution.

Weekly Cadmium Intake	Area A (*n* = 712)	Area B (*n* = 432)
Median	7.2	6.0 *
25–75th percentiles	5.2–9.7	4.4–8.5
Ranges	1.5–42	1.3–26
<7 µg/kg/week	344 (48.3%)	268 (62.0%)
≥7 µg/kg/week	368 (51.7%)	164 (38.0%)

*: *p* < 0.05 versus area A (median test).

**Table 14 toxics-08-00044-t014:** Age-classified weekly cadmium intake per body weight (µg/kg BW/week) in female farmers in areas A and B.

Age Groups (Mean Age ± SD)	*n*	Median	25–75th Percentile
Area A (*n* = 712)			
20–29 years (25.0 ± 3.0)	27	3.6	2.4–5.9
30–39 years (35.0 ± 3.2)	27	4.8	3.5–7.2
40–49 years (45.4 ± 2.9)	109	6.3 *	4.5–8.7
50–59 years (54.6 ± 2.9)	213	7.1 *	5.4–9.5
60–69 years (64.6 ± 2.8)	278	8.1 *	6.0–11
70 years– (72.8 ± 2.2)	58	7.8 *	6.0–11
Area B (*n* = 432)			
30–39 years (36.8 ± 1.5)	14	6.3	4.5–14
40–49 years (45.8 ± 2.7)	85	5.1	3.2–7.7
50–59 years (54.6 ± 2.8)	151	5.7	3.9–7.5
60–69 years (63.9 ± 2.8)	143	6.8	5.2–9.0
70 years– (73.1 ± 2.8)	39	6.6	5.5–11

*: *p* < 0.05 versus 20–29 years (Steel-Dwass test).

**Table 15 toxics-08-00044-t015:** Results of the Monte Carlo simulation for weekly cadmium intake per body weight (µg/kg BW/week) in female farmers in areas A and B.

Weekly Cadmium Intake	Area A (*n* = 710) *	Area B (*n* = 432)
Median	7.0	6.0
5–95th percentiles	3.1–17.0	2.6–15.9
Ranges	1.2–70.9	0.9–103

*: Two outliers were excluded.

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
