# Peer review of "Exposure Assessment of Cadmium in Female Farmers in Cadmium-Polluted Areas in Northern Japan"

_toxics, 2020, doi:10.3390/toxics8020044_

Round 1
Reviewer 1 Report
In my eyes, this paper should not be accepted for publication in Contaminants journal by the following reason:
The study uses data very old. The results obtained are not current. Why did the authors use such an old food consumption survey? These dates were provided in 2001 – 2002 and this paper has been written in 2020. Furthermore, products were purchased in 2003. Almost 20 years have passed.
Secondly, the results are scarcely discussed. Why did not the authors compare concentration data in food with previous studies? There is an abundant bibliography on this topic. Furthermore, the only referenced authors have the same nationality as the authors. Please, provide other references from European and American researches.
Finally, results are expressed and discussed in a redundant manner. The same content could have been said in a shorter and clearer way
Other relevant considerations:
Lines 41 – 44. What is the reason because Toyama area has a high Cd concentration? Why Toyama area was completely restored in 2011?
I suggest authors create a map of Japan showing the areas in which the study was developed. In this map the location of mining areas and their affiliated smelters, rivers should be indicated.
Lines 71 – 73. This latter sentence must be moved to Conclusions section
Lines 90 – 111. Certified Reference Materials (CRM) must be provided in order to show the accuracy of obtained results
Line 221. “There were statistically significant differences in height and weight” Please, provide p value.
Lines 236 – 243. I recommend authors move this paragraph to Discussion section
Author Response
Responses to Reviewer 1’s Comments
Thank you very much for your valuable comments, according to which we revised the manuscript. Point-by-point responses are provided below. The parts that we changed are highlighted in the manuscript.
Point 1: The study uses data very old. The results obtained are not current. Why did the authors use such an old food consumption survey? These dates were provided in 2001 – 2002 and this paper has been written in 2020. Furthermore, products were purchased in 2003. Almost 20 years have passed.
Response 1: The purpose of the present study was not to show the current situation in Cd-polluted areas in Akita, but to examine previous exposure to high Cd levels in local farmers. We did not submit the data collected at that time for publication. Nevertheless, the results obtained remain important, even after approximately 20 years, because few Cd-polluted areas and subjects exposed to high levels of Cd remain in Japan. Furthermore, data were collected at the start of the flooding of paddy fields, which has successfully decreased Cd exposure levels in farmers in these areas; therefore, these data will never be obtained again in the future.
We added the following information to the Discussion, “Since the data used were obtained in 2001–2003, the current situation in these Cd-polluted areas remains unclear. Nevertheless, the results obtained remain important, even after approximately 20 years, because few Cd-polluted areas remain in Japan. Furthermore, data were collected at the start of the flooding of paddy fields, which has successfully decreased Cd exposure levels in farmers in these areas; therefore, these data will never be obtained again in the future.” on lines 410–415.
Point 2: Secondly, the results are scarcely discussed. Why did not the authors compare concentration data in food with previous studies? There is an abundant bibliography on this topic. Furthermore, the only referenced authors have the same nationality as the authors. Please, provide other references from European and American researches.
Response 2: We added the following information to the Discussion, “The National Institute of Health Sciences previously investigated Cd intake by the general Japanese population using the market basket method [15], and an assessment of Cd intake based on the individual food analysis, similar to the present study, was not previously performed in Japan. Cd intake in the Jinzu River basin of Toyama, the heaviest Cd-polluted area in Japan, was reported to be as high as 600 µg/day in 1968 using the market basket method [19]. On the other hand, in Kosaka town in Akita, the intake of Cd from rice by local residents, assessed from Cd concentrations in rice, was 100.6 µg/day in 1974–1976 [20], and 92 µg/day in 1978 and 55 µg/day in 1999-2000 using the duplicate portion study [19]; the latter was similar to the present results. These findings indicate that Cd intake in Cd-polluted areas in Akita has been decreasing for decades, and are consistent with the present results. Cd intake in Cd-polluted areas in Akita are still higher than those by the general Japanese population and Western counties: Recent Cd intake by general populations assessed using the market basket method was 4.63 μg/day or 0.54 μg/kg body weight/week in U.S.A. [21], 0.16 μg/kg body weight/week in France [22], 0.77 μg/kg body weight/week in Spain [23], 1 μg/kg body weight/week in Sweden [24], 5.00 μg/day in Italy [25], 0.85μg/kg body weight/week in Belgium [26], and 13.5 μg/day in Denmark [27]. Although Cd intake in Eastern Asian countries, where rice is consumed as a staple food, is generally high, such as 32.7 μg/day in China [28] and 22.0 μg/day in Korea [29], Cd intake levels remain higher in Akita.” on lines 393–409. We also added references [19-29].
Furthermore, Itoh et al. (Int J Hyg Environ Health. 2014) examined Cd intake by Japanese individuals, but used data on Cd concentrations in food items from the Japanese government, which is similar to the results shown in Figure 2.
Point 3: Finally, results are expressed and discussed in a redundant manner. The same content could have been said in a shorter and clearer way.
Response 3: Although this comment does not specify the changes needed, we modified the parts described below.
We changed “Median total Cd intake levels in areas A and B were 55.7 and 47.8 µg/day, respectively, with the former being significantly higher than the latter. This result was attributed to the significant difference in Cd intake in the subgroup of rice and rice products because the Cd intake levels in other subgroups were all similar between the 2 areas (Table 3).” to “Cd intake from the subgroup of rice and rice products was significantly higher in area A than in area B, while those from other subgroups were similar between the 2 areas, except for vegetables, with Cd intake being significantly lower in area A than in area B. Median total Cd intake levels in areas A and B were 55.7 and 47.8 µg/day, respectively, with the former being significantly higher than the latter.” on lines 247–251.
We deleted “These results confirmed that the estimation of Cd exposure in our previous study was appropriate and that high Cd exposure levels, as confirmed by the high blood and urinary Cd levels of our subjects, were derived from high Cd intakes through local food items.” from the first paragraph of the Discussion.
We changed “The intake of Cd was higher in area A than in area B, which was attributed to differences in rice Cd concentrations. However,” to “Although the intake of Cd was higher in area A than in area B,” on line 319.
Point 4: Lines 41 – 44. What is the reason because Toyama area has a high Cd concentration? Why Toyama area was completely restored in 2011?
Response 4: Since itai-itai disease is already well known, we did not add a detailed explanation. However, we changed “A large number of patients with Cd nephropathy were identified in the heaviest Cd-polluted area along the Jinzu River basin of Toyama prefecture in Japan, with 200 patients with itai-itai disease being officially recognized there by 2020.” to “The heaviest Cd-polluted area in Japan was along the Jinzu River basin of Toyama prefecture, at which agricultural fields were contaminated by a large amount of Cd derived from an upstream mine. A large number of inhabitants in this area developed Cd nephropathy, and 200 patients with itai-itai disease were officially recognized by 2020.” on lines 40–44.
Point 5: I suggest authors create a map of Japan showing the areas in which the study was developed. In this map the location of mining areas and their affiliated smelters, rivers should be indicated.
Response 5: We added a map and an accompanying legend as Figure 1. Therefore, the numbers of figures have changed.
Point 6: Lines 71 – 73. This latter sentence must be moved to Conclusions section.
Response 6: According to the Instructions for Authors of Toxics, “Finally, briefly mention the main aim of the work and highlight the main conclusions.” in Introduction. Therefore, we briefly described the main conclusions here.
Point 7: Lines 90 – 111. Certified Reference Materials (CRM) must be provided in order to show the accuracy of obtained results.
Response 7: According to the Japan Food Research Laboratories, at which Cd measurements were performed, quality control was achieved using sugar (commercial products) added with Cd as an alternative to certified reference materials. Its additional recovery was maintained within 90-110%. We measured Cd concentrations in 100 types of food items, and CRMs were not available for all of them.
Therefore, we added “Quality control was achieved in the analysis using sugar (commercial products) added with Cd as an alternative to certified reference materials. Its additional recovery was maintained within 90-110%.” on lines 120–122.
Point 8: Line 221. “There were statistically significant differences in height and weight” Please, provide p value.
Response 8: We added “(p=0.047)” and “(p=0.047)”, respectively.
We added mean rice intake in Table 2 in order to further demonstrate that the background of the subjects in areas A and B did not significantly differ. Furthermore, we changed “in age or energy intake” to “in age, energy intake, or rice intake” on line 238, and “and total energy intake” to “total energy intake, and rice intake” on line 180.
Point 9: Lines 236 – 243. I recommend authors move this paragraph to Discussion section.
Response 9: We do not agree with this recommendation. We consider it important to demonstrate that the calculated Cd intake levels in Akita were high and not just show the calculated results.
# Other corrected parts:
We changed “Cadmium-polluted areas exist in Akita prefecture, located in the northern part of Japan.” to “Akita prefecture, located in the northern part of Japan, has many cadmium-polluted areas.”on lines 19–20.
We changed “We performed an exposure assessment of cadmium on 712 and 432 female farmers” to “We herein performed an exposure assessment of cadmium in 712 and 432 female farmers” on line 20.
We changed “of” to “reported by” on line 28.
We changed “of Japan” to “in Japan” on line 29.
We changed “there are still large, but scattered, Cd-polluted areas in Akita prefecture, located in the northern part of Japan, due to the previous activities of mines and smelters” to “large, but scattered, Cd-polluted areas still remain in Akita prefecture, which is located in the northern part of Japan, due to the previous activities of mines and smelters” on lines 45–47.
We changed “and” to “while” on line 53.
We changed “of Cd on subjects on Cd in subjects” to “” on line 62.
We changed “the inhabitants” to “its inhabitants” on line 68.
We changed “verify” to “confirm” on line 77.
We changed “Labor” to “Labour” on line 89.
We changed “in the diet surveys” to “in diet surveys” on line 90.
We changed “the Japan Food Research Laboratories” to “Japan Food Research Laboratories” on lines 104–105.
We changed “In total, 0.4 and 0.8 µg/mL standard Cd solutions were made by diluting the original standard Cd solution (Kanto Chemical Co., Inc., Tokyo, Japan) with 1% hydrochloric acid (HCl).” to “” on lines 118–120.
We changed “food” to “food items” on lines 112–113.
We changed “and” to “while” on line 149.
We changed “row” to “raw” on line 156.
We changed “The intakes of mushrooms were” to “The intake of mushrooms was” on line 157.
We changed “cod” to “of cod” on line 160.
We changed “by DHQ” to “in DHQ on line 161.
We changed “Cd concentrations” to “the Cd concentrations” on line 161.
We changed “scallop” to “scallops” on line 162.
We changed “to their corresponding food intakes” to “by their corresponding food intake” on line 170.
We changed “of individual subjects” to “in individual subjects” on lines 170– 171.
We changed “Cd concentrations” to “the Cd concentration” on line 175.
We changed “the standard tables of food compositions” to “the Standard Tables of Food Compositions” on line 177.
We changed “that their distributions were log-normal” to “of a lognormal distribution” on line 191.
We changed “These statistical analyses” to “Statistical analyses” on line 192.
We changed “Lists of food items and their Cd concentrations, divided into 10 subgroups, are shown in Table 1.” to “Food items and their Cd concentrations, divided into 10 subgroups, are listed in Table 1.” on line 195.
We changed “less” to “lower” on line 197.
We changed “, but 8.2% and 5.8% of Cd concentrations in rice were over the safety standard in areas A and B, respectively” to “however, Cd concentrations of 8.2 and 5.8% in rice in areas A and B, respectively, were above the safety standard” on lines 197–198.
We changed “is” to “are” on line 212.
We changed “Dropworta” to “Dropwort” in table 1-d.
We changed “. There were statistically significant differences” to “, whereas significant differences were noted” on lines 238–239.
We changed “between them, but the differences are very small and biologically negligible” to “, but were biologically negligible.” on line 239.
We changed “were” to “was” on line 242.
We changed “in” to “from” on line 244.
We changed “total Cd intake” to “he total Cd intake” on lines 245–246.
We changed “in” to “from” on line 246.
We changed “7.2 µg/kg BW/week in area A and 6.0 µg/kg BW/week in area B, both of which were approximately PTWI or TWI” to “7.2 and 6.0 µg/kg BW/week in areas A and B, respectively, with both being approximately PTWI or TWI” on lines 272–273.
We changed “distribution” to “distributions” on line 273.
We changed “If data obtained from 2 subjects in area A that were extremely large and considered to be outliers are excluded, the distributions in both areas became similar and skewed to the higher side.” to “The exclusion of data obtained from 2 subjects in area A that were extremely large and considered to be outliers resulted in similar distributions in both areas that skewed to the higher side.” on lines 274–276.
We changed “Cd intakes, 7.0 and 6.0” to “Cd intake levels of 7.0 and 6.0 µg/kg BW/week” on lines 294–295.
We changed “very close” to “similar” on line 295.
We changed “Cd concentrations” to “their Cd concentrations” on line 306.
We changed “high levels of Cd” to “high Cd levels” on line 307.
We changed “The results obtained” to “The present results” on line 308.
We changed “Cd intakes were” to “Cd intake was” on line 309.
We changed “by our subjects than by the general population” to “in our subjects than in the general population” on line 309.
We changed “weekly Cd intakes per body weight, which were verified by the Monte Carlo simulation, were approximately the same levels of the PTWI of JECFA or TWI of Japan, and 38.0–51.7% of the subjects’ weekly Cd intakes per body weight were over it” to “weekly Cd intake per body weight, which was confirmed by the Monte Carlo simulation, were approximately the same as the PTWI of JECFA or TWI of Japan, while 38.0–51.7% of subjects had weekly Cd intake per body weight that was above it” on lines 310–312.
We changed “there were rice with Cd concentrations over the safety standard” to “rice also had Cd concentrations that were above the safety standard” on lines 317–318.
We changed “were” to “was” on line 334.
We changed “were” to “was” on line 335.
We changed “were” to “was” on line 337.
We changed “in general Japanese” to “the general Japanese population” on line 338.
We changed “were” to “was” on line 338.
We changed “in general Japanese” to “the general Japanese population” on line 342.
We changed “Cd concentrations in sesame seeds, taro (satoimo in Japanese), yams (yamaimo in Japanese), soybeans, carrot, spinach, burdock (gobo in Japanese), garland chrysanthemum (shungiku in Japanese), Japanese mustard spinach (komatsuna in Japanese), garlic, belvedere fruit (tomburi in Japanese), and shiitake mushroom were high” to “Cd concentrations were high in sesame seeds, taro (satoimo in Japanese), yams (yamaimo in Japanese), soybeans, carrot, spinach, burdock (gobo in Japanese), garland chrysanthemum (shungiku in Japanese), Japanese mustard spinach (komatsuna in Japanese), garlic, belvedere fruit (tomburi in Japanese), and shiitake mushroom” on lines 349–352.
We changed “Cd concentrations in livestock food and fruit were generally low” to “Cd concentrations were generally low in livestock food and fruit” on line 356.
We changed “Cd concentrations in tubers, soybeans, brightly colored vegetables, and seafood, particularly seaweed, and the innards of squid or shellfish were high in Japan” to “Cd concentrations were high in tubers, soybeans, brightly colored vegetables, and seafood, particularly seaweed, and the innards of squid or shellfish in Japan” on lines 357–359.
We changed “The median weekly Cd intakes” to “Median weekly Cd intake levels” on line 360.
We changed “the chemical concentrations in individual foodstuffs remain unknown” to “the concentration of this chemical in individual food items remains unknown” on lines 378–379.
We changed “of individuals” to “by individuals” on line 380.
We changed “the present study” to “the present results” on line 382.
We changed “In general, the individual food analysis” to “The individual food analysis generally” on line 386.
We changed “In addition” to “Furthermore” on line 388.
We changed “the standard tables of food compositions” to “the Standard Tables of Food Compositions” on lines 389–390.
We changed “The subjects” to “Furthermore, the subjects” on line 416.
We changed “remain” to “remains” on line 417.
We changed “do not reflect” to “did not reflect” on line 421.
We changed “on female farmers” to “in female farmers” on lines 431.

Reviewer 2 Report
This manuscript, detailing the effect of Cd consumption in female Japanese farmers, addresses an important topic of exposure assessment. It also seems to be a follow up of the author's work. For that reason I believe it can be improved by marking more clearly the novelty of this study. Bellow I have summarised the questions I had while reading the manuscript, which I hope the authors find useful for the revision.
Abstract
Line 28 – do not use acronyms such as PTWI, TWI and CODEX before explaining in full what they mean
I could not find the CODEX listed in the References. I wasn't sure if you meant ref [7], but in any case if you are referring to the WHO document as "CODEX" that information must be given.
Introduction
Line 55 – Similar to the previous comment: what does JECFA stands for?
Line 58 to 64 - you explain the reason why you believe a new assessment is necessary for area A but not for area B. Nevertheless you proceed in Line 65 to state that a study is needed in areas A and B. Please include a reasoning for area B as well, when detailing the work previously done.
Overall, at the end of the Introduction I was not able to understand what work had already been performed and what was the novelty of the work performed in the present paper. I was left with the impression that all the data had already been collected and analysed and that the Monte Carlo simulation applied to that data was the only difference from this paper to previous works.
This comment also refers to the Material and Methods section 2.1 to 2.4. How much of this work was performed previously, and how much is new should be clearly stated.
I couldn't also understand the Cd measurement in rice. In Line 110 it seems that Cd was measured externally, however, in Line 116 it seems that you performed some of those measurements?
You should include the analytical technique for the rice / blood / urine samples analysed, and a reference that shows those methods have been validated. These biological matrixes are very diverse and, therefore, it is essential that you show that you used validated protocols.
Line 164 – There's a reference missing
Section 2.5
I couldn't understand why you chose GMs for the Cd results. They didn't follow a normal distribution?
Results
Table 3 – I couldn't understand why the values for all the groups in Area B were significant (p<0.05). It also seems to contradict the sentence in Line 231, in which you state that the only significant difference was in the rice and rice products.
Line 236 to 243 – I couldn't understand the values you are referring to: 31.1 µg/day (1981) and etc. are not show in Fig 1. And I can not understand what the different values refer to, for example, 21.1 µg/day in 2007 (Line 238) and 7.8 µg/day in 2007 (Line 240).
Figure 1 – I couldn't understand what µg/man/day stands for. What is the man?
If I understood right you are showing median values for the Japanese population and averages for Area A and B. You must decide on one measure: either median or average and present all the values measured in the same way.
Line 253 – What were the excluded outliers? Notice the spelling: outliers. You should correct it throughout the manuscript
Discussion
Line 282 – You do not show this data on the histograms that you present. If you wish to draw attention to this data you should consider presenting it in some other way.
Line 282 – This is the first time we hear about the results on blood and urine. If these are part of this work you should analyse it in the Results section, otherwise you should give the reference in which these values were obtained.
Line 291 – You state that "accumulated levels of Cd in subjects were higher in area B than in area A" but in subsequent lines you contradict this "9.34 and 4.90 μg/g cr. in areas A and B, respectively".
Line 297 – If the "flooding of paddy fields" seems to be the explanation for these discrepancies it is not clear to the reader why you didn't check that in the present work, comparing it to the 2002-2003 values.
Line 340 – This is the first time we hear about the 2 TSD approaches. If these are part of this work you should analyse it in the Results section, otherwise you should give the reference in which these values were obtained.
Author Response
Responses to Reviewer 2’s Comments
Thank you very much for your valuable comments, according to which we revised the manuscript. Point-by-point responses are provided below. The parts that we changed are highlighted in the manuscript.
Point 1: Line 28 – do not use acronyms such as PTWI, TWI and CODEX before explaining in full what they mean.
Response 1: In the Abstract, we changed “PTWI” to “the provisional tolerable weekly intake” and “TWI” to “the tolerable weekly intake”.
“CODEX” is a proper noun, not an acronym, which has a meaning “Food Code”.
However, since “CODEX” is not accurate and “JECFA” is appropriate, we changed “CODEX” to “the Joint FAO/WHO Expert Committee on Food Additives”.
Point 2: I could not find the CODEX listed in the References. I wasn't sure if you meant ref [7], but in any case if you are referring to the WHO document as "CODEX" that information must be given.
Response 2: “CODEX” is not accurate and “JECFA” is appropriate. Therefore, we deleted “of CODEX”, and replaced “CODEX” with “JECFA” throughout the manuscript.
Reference [7] is a report by JECFA.
Point 3: Line 55 – Similar to the previous comment: what does JECFA stands for?
Response 3: We added an explanation of “JECFA” on lines 57–58.
Point 4: Line 58 to 64 - you explain the reason why you believe a new assessment is necessary for area A but not for area B. Nevertheless you proceed in Line 65 to state that a study is needed in areas A and B. Please include a reasoning for area B as well, when detailing the work previously done.
Response 4: We changed “in areas A and B” to “in this area” on line 68, and added “A similar exposure assessment of Cd in area B, in which Cd pollution was heavier than in area A, needs to be conducted.” on lines 68–70.
Point 5: Overall, at the end of the Introduction I was not able to understand what work had already been performed and what was the novelty of the work performed in the present paper. I was left with the impression that all the data had already been collected and analysed and that the Monte Carlo simulation applied to that data was the only difference from this paper to previous works.
Response 5: We added “previously” to the Abstract and changed “we assessed Cd exposure levels in 2 Cd-polluted areas in Akita by an individual food analysis method using Cd concentrations in individual local food items and their intake amounts by subjects undergoing local health examinations” to “we newly collected 100 types of local food items in 2 Cd-polluted areas in Akita, measured their Cd concentrations, and assessed Cd exposure levels in the subjects of previous local health examinations using an individual food analysis method based on Cd concentrations in individual food items and the intake amounts of these food items by subjects obtained from diet surveys conducted at health examinations.” on lines 71–75.
Point 6: This comment also refers to the Material and Methods section 2.1 to 2.4. How much of this work was performed previously, and how much is new should be clearly stated.
Response 6: We changed “Health examinations were performed in areas A and B in 2001-2004 on 725 and 438 females, respectively; 12 and 6 with extremely low or high energy (≤1000 or ≥3500 kcal/day) were excluded in areas A and B, respectively, and one whose consumption of rice was zero in area A, resulting in 712 and 432 subjects for analyses. In health examinations, we collected self-harvested rice from each subject to measure Cd concentrations and also obtained blood and urine specimens. Subjects were asked to complete diet surveys using a diet history questionnaire (DHQ), which is designed to assess food and nutrient intake levels in the previous month based on the quantity and semiquantitative frequency of consumption of 110 food items commonly eaten in Japan [11]. Estimates of intake for food, energy, and selected nutrients were calculated using an ad hoc computer algorithm for DHQ based on Standard Tables of Food Composition in Japan, which has already been validated [12,13]. We used data obtained on individual food intake to calculate Cd intake.” to “Health examinations performed in areas A and B in 2001-2004 were described previously [4.5]. In the present study, we used data obtained on age, height, weight, and the results of a diet history questionnaire (DHQ) from 725 and 438 subjects in areas A and B, respectively. DHQ is designed to assess food and nutrient intake levels in the previous month based on the quantity and semiquantitative frequency of the consumption of 110 food items commonly eaten in Japan [11]. Estimates of intake for food, energy, and selected nutrients were calculated using an ad hoc computer algorithm for DHQ based on Standard Tables of Food Composition in Japan, which has already been validated [12,13]. We used data obtained on individual food intake to calculate Cd intake. Among subjects, 12 and 6 with extremely low or high energy intake (≤1000 or ≥3500 kcal/day) were excluded in areas A and B, respectively, in addition to one whose consumption of rice was zero in area A, resulting in 712 and 432 subjects for analyses.” on lines 125–135.
Point 7: I couldn't also understand the Cd measurement in rice. In Line 110 it seems that Cd was measured externally, however, in Line 116 it seems that you performed some of those measurements?
Response 7: All Cd concentrations in rice were measured by IDEA Consultants, Inc.
Point 8: You should include the analytical technique for the rice / blood / urine samples analysed, and a reference that shows those methods have been validated. These biological matrixes are very diverse and, therefore, it is essential that you show that you used validated protocols.
Response 8: In the present study, we did not obtain data on blood or urine Cd concentrations and did not measure Cd concentrations in rice. Therefore, it may be misleading to include methods that describe their measurement in the Materials and Methods.
Therefore, we deleted “Separate from the measurement of Cd concentrations in the food items collected, individual Cd concentrations in rice obtained in previous health examinations, which were measured by IDEA Consultants, Inc. (Metocean Environment, Inc.) (Shizuoka, Japan), were used to calculate Cd intake from rice [4,5].”, and added “, which were obtained in previous health examinations,” on line 171.
Point 9: Line 164 – There's a reference missing.
Response 9: We added the reference “Masironi, R.; Koirtyohann, S.R.; Pierce, J.O. Zinc, copper, cadmium and chromium in polished and unpolished rice. Sci. Total Environ. 1977, 7, 27–43.” as no. 14. Therefore, the numbers of references were changed.
Point 10: Section 2.5: I couldn't understand why you chose GMs for the Cd results. They didn't follow a normal distribution?
Response 10: Since Cd concentrations follow a clear lognormal distribution because of large numbers, and the distribution of Cd concentrations in other foodstuffs was not clear because of small numbers, we considered it appropriate to use GM for rice and AM for other foodstuffs to calculate Cd intake.
Therefore, we changed “Cd concentrations in rice were presented as GMs with 25th and 75th percentiles.” to “Since Cd concentrations in rice followed a clear lognormal distribution with large numbers and the distribution of Cd concentrations in other food items was not clear because of small numbers, rice and other food items were shown as GMs and AMs, respectively.” on lines 182–185.
Point 11: Results: Table 3 – I couldn't understand why the values for all the groups in Area B were significant (p<0.05). It also seems to contradict the sentence in Line 231, in which you state that the only significant difference was in the rice and rice products.
Response 11: We corrected Table 3 and its explanation “Median total Cd in areas A and B were 55.7 and 47.8 µg/day, respectively, with the former being significantly higher than the latter. This result was attributed to the significant difference in Cd intake in the subgroup of rice and rice products because the Cd in other subgroups were all similar between the 2 areas (Table 3).” to “Cd intake from the subgroup of rice and rice products was significantly higher in area A than in area B, while those from other subgroups were similar between the 2 areas, except for vegetables, with Cd intake being significantly lower in area A than in area B. Median total Cd in areas A and B were 55.7 and 47.8 µg/day, respectively, with the former being significantly higher than the latter.” on lines 247–251.
Point 12: Line 236 to 243 – I couldn't understand the values you are referring to: 31.1 µg/day (1981) and etc. are not show in Fig 1. And I can not understand what the different values refer to, for example, 21.1 µg/day in 2007 (Line 238) and 7.8 µg/day in 2007 (Line 240).
Response 12: We changed “Cd intakes have been gradually decreasing in Japan: 31.1 µg/day in 1981, 21.1 µg/day in 2007, and 17.8 µg/day in 2015, which is mainly due to the decrease in Cd intake from rice.” to “Cd intake has been gradually decreasing in Japan: 31.1 µg/day (16.2 µg/day from rice and 14.9 µg/day from other food items) in 1981, 21.1 µg/day (7.8 µg/day from rice and 13.3 µg/day from other food items) in 2007, and 17.8 µg/day (5.7 µg/day from rice and 12.1 µg/day from other food items) in 2015, and this has been mainly attributed to a reduction in Cd intake from rice.” on lines 256–259.
Point 13: Figure 1 – I couldn't understand what µg/man/day stands for. What is the man?
Response 13: We changed “µg/man/day” to “µg/day” in Figure 2 and added “per person” in the legend. We also added “per person” to Table 3.
Point 14: If I understood right you are showing median values for the Japanese population and averages for Area A and B. You must decide on one measure: either median or average and present all the values measured in the same way.
Response 14: Cd intake values in the general Japanese population were shown as averages, while those in Cd-polluted areas A and B were provided as medians. The official announcement from the Japanese government only provides these values as “averages”. In contrast, our data on Cd intake in Cd-polluted areas followed a lognormal distribution. Therefore, it is impossible and inappropriate to use them as averages or medians alone.
We added “The Japanese government provides Cd intake values by the general population as averages. Although it may be statistically inappropriate to compare averages in the general population with medians in the Cd-polluted areas, it would not be a serious problem because of the large difference between them.” as one of the limitations on lines 426–429.
Point 15: Line 253 – What were the excluded outliers? Notice the spelling: outliers. You should correct it throughout the manuscript.
Response 15: We corrected “outliners” to “outliers” on lines 189, 190, 275, 293, 303 (Table 6).
Point 16: Line 282 – You do not show this data on the histograms that you present. If you wish to draw attention to this data you should consider presenting it in some other way.
Response 16: We added its distribution and “and their distribution” to Table 4, and “(p<0.05, χ2 test) (Table 4)” on line 279. We also added “The χ2 test was used to compare the proportion of Cd intake per body weight above the PTWI or TWI.” to Statistical analysis on lines 186–187. Furthermore, we changed “40–50%” to “38.0–51.7%” on lines 311–312.
Point 17: Line 282 – This is the first time we hear about the results on blood and urine. If these are part of this work you should analyse it in the Results section, otherwise you should give the reference in which these values were obtained.
Response 17: We deleted these parts.
Point 18: Line 291 – You state that "accumulated levels of Cd in subjects were higher in area B than in area A" but in subsequent lines you contradict this "9.34 and 4.90 μg/g cr. in areas A and B, respectively".
Response 18: We corrected “9.34 and 4.90 µg/g cr.” to “4.90 and 9.34 µg/g cr.” on line 321.
Point 19: Line 297 – If the "flooding of paddy fields" seems to be the explanation for these discrepancies it is not clear to the reader why you didn't check that in the present work, comparing it to the 2002-2003 values.
Response 19: We do not understand the meaning of this comment. We stated that the flooding of paddy fields started in 2002, and we compared Cd intake levels between area A in 2001–2002 and area B in 2002–2003.
Point 20: Line 340 – This is the first time we hear about the 2 TSD approaches. If these are part of this work you should analyse it in the Results section, otherwise you should give the reference in which these values were obtained.
Response 20: We performed a duplicate portion study in area A, but did not include the data obtained. Therefore, we added “(unpublished data)” on line 383.
# Other corrected parts:
We changed “Cadmium-polluted areas exist in Akita prefecture, located in the northern part of Japan.” to “Akita prefecture, located in the northern part of Japan, has many cadmium-polluted areas.”on lines 19–20.
We changed “We performed an exposure assessment of cadmium on 712 and 432 female farmers” to “We herein performed an exposure assessment of cadmium in 712 and 432 female farmers” on line 20.
We changed “of” to “reported by” on line 28.
We changed “of Japan” to “in Japan” on line 29.
We changed “there are still large, but scattered, Cd-polluted areas in Akita prefecture, located in the northern part of Japan, due to the previous activities of mines and smelters” to “large, but scattered, Cd-polluted areas still remain in Akita prefecture, which is located in the northern part of Japan, due to the previous activities of mines and smelters” on lines 45–47.
We changed “and” to “while” on line 53.
We changed “of Cd on subjects on Cd in subjects” to “” on line 62.
We changed “the inhabitants” to “its inhabitants” on line 68.
We changed “verify” to “confirm” on line 77.
We changed “Labor” to “Labour” on line 89.
We changed “in the diet surveys” to “in diet surveys” on line 90.
We changed “the Japan Food Research Laboratories” to “Japan Food Research Laboratories” on lines 104–105.
We changed “In total, 0.4 and 0.8 µg/mL standard Cd solutions were made by diluting the original standard Cd solution (Kanto Chemical Co., Inc., Tokyo, Japan) with 1% hydrochloric acid (HCl).” to “” on lines 118–120.
We changed “food” to “food items” on lines 112–113.
We changed “and” to “while” on line 149.
We changed “row” to “raw” on line 156.
We changed “The intakes of mushrooms were” to “The intake of mushrooms was” on line 157.
We changed “cod” to “of cod” on line 160.
We changed “by DHQ” to “in DHQ on line 161.
We changed “Cd concentrations” to “the Cd concentrations” on line 161.
We changed “scallop” to “scallops” on line 162.
We changed “to their corresponding food intakes” to “by their corresponding food intake” on line 170.
We changed “of individual subjects” to “in individual subjects” on lines 170– 171.
We changed “Cd concentrations” to “the Cd concentration” on line 175.
We changed “the standard tables of food compositions” to “the Standard Tables of Food Compositions” on line 177.
We changed “that their distributions were log-normal” to “of a lognormal distribution” on line 191.
We changed “These statistical analyses” to “Statistical analyses” on line 192.
We changed “Lists of food items and their Cd concentrations, divided into 10 subgroups, are shown in Table 1.” to “Food items and their Cd concentrations, divided into 10 subgroups, are listed in Table 1.” on line 195.
We changed “less” to “lower” on line 197.
We changed “, but 8.2% and 5.8% of Cd concentrations in rice were over the safety standard in areas A and B, respectively” to “however, Cd concentrations of 8.2 and 5.8% in rice in areas A and B, respectively, were above the safety standard” on lines 197–198.
We changed “is” to “are” on line 212.
We changed “Dropworta” to “Dropwort” in table 1-d.
We changed “. There were statistically significant differences” to “, whereas significant differences were noted” on lines 238–239.
We changed “between them, but the differences are very small and biologically negligible” to “, but were biologically negligible.” on line 239.
We changed “were” to “was” on line 242.
We changed “in” to “from” on line 244.
We changed “total Cd intake” to “he total Cd intake” on lines 245–246.
We changed “in” to “from” on line 246.
We changed “7.2 µg/kg BW/week in area A and 6.0 µg/kg BW/week in area B, both of which were approximately PTWI or TWI” to “7.2 and 6.0 µg/kg BW/week in areas A and B, respectively, with both being approximately PTWI or TWI” on lines 272–273.
We changed “distribution” to “distributions” on line 273.
We changed “If data obtained from 2 subjects in area A that were extremely large and considered to be outliers are excluded, the distributions in both areas became similar and skewed to the higher side.” to “The exclusion of data obtained from 2 subjects in area A that were extremely large and considered to be outliers resulted in similar distributions in both areas that skewed to the higher side.” on lines 274–276.
We changed “Cd intakes, 7.0 and 6.0” to “Cd intake levels of 7.0 and 6.0 µg/kg BW/week” on lines 294–295.
We changed “very close” to “similar” on line 295.
We changed “Cd concentrations” to “their Cd concentrations” on line 306.
We changed “high levels of Cd” to “high Cd levels” on line 307.
We changed “The results obtained” to “The present results” on line 308.
We changed “Cd intakes were” to “Cd intake was” on line 309.
We changed “by our subjects than by the general population” to “in our subjects than in the general population” on line 309.
We changed “weekly Cd intakes per body weight, which were verified by the Monte Carlo simulation, were approximately the same levels of the PTWI of JECFA or TWI of Japan, and 38.0–51.7% of the subjects’ weekly Cd intakes per body weight were over it” to “weekly Cd intake per body weight, which was confirmed by the Monte Carlo simulation, were approximately the same as the PTWI of JECFA or TWI of Japan, while 38.0–51.7% of subjects had weekly Cd intake per body weight that was above it” on lines 310–312.
We changed “there were rice with Cd concentrations over the safety standard” to “rice also had Cd concentrations that were above the safety standard” on lines 317–318.
We changed “were” to “was” on line 334.
We changed “were” to “was” on line 335.
We changed “were” to “was” on line 337.
We changed “in general Japanese” to “the general Japanese population” on line 338.
We changed “were” to “was” on line 338.
We changed “in general Japanese” to “the general Japanese population” on line 342.
We changed “Cd concentrations in sesame seeds, taro (satoimo in Japanese), yams (yamaimo in Japanese), soybeans, carrot, spinach, burdock (gobo in Japanese), garland chrysanthemum (shungiku in Japanese), Japanese mustard spinach (komatsuna in Japanese), garlic, belvedere fruit (tomburi in Japanese), and shiitake mushroom were high” to “Cd concentrations were high in sesame seeds, taro (satoimo in Japanese), yams (yamaimo in Japanese), soybeans, carrot, spinach, burdock (gobo in Japanese), garland chrysanthemum (shungiku in Japanese), Japanese mustard spinach (komatsuna in Japanese), garlic, belvedere fruit (tomburi in Japanese), and shiitake mushroom” on lines 349–352.
We changed “Cd concentrations in livestock food and fruit were generally low” to “Cd concentrations were generally low in livestock food and fruit” on line 356.
We changed “Cd concentrations in tubers, soybeans, brightly colored vegetables, and seafood, particularly seaweed, and the innards of squid or shellfish were high in Japan” to “Cd concentrations were high in tubers, soybeans, brightly colored vegetables, and seafood, particularly seaweed, and the innards of squid or shellfish in Japan” on lines 357–359.
We changed “The median weekly Cd intakes” to “Median weekly Cd intake levels” on line 360.
We changed “the chemical concentrations in individual foodstuffs remain unknown” to “the concentration of this chemical in individual food items remains unknown” on lines 378–379.
We changed “of individuals” to “by individuals” on line 380.
We changed “the present study” to “the present results” on line 382.
We changed “In general, the individual food analysis” to “The individual food analysis generally” on line 386.
We changed “In addition” to “Furthermore” on line 388.
We changed “the standard tables of food compositions” to “the Standard Tables of Food Compositions” on lines 389–390.
We changed “The subjects” to “Furthermore, the subjects” on line 416.
We changed “remain” to “remains” on line 417.
We changed “do not reflect” to “did not reflect” on line 421.
We changed “on female farmers” to “in female farmers” on lines 431.

Reviewer 3 Report
In this manuscript, the authors reconsider food samples obtained in the early ‘00s to estimate the exposure to cadmium of populations that were studied then, with results published long ago (refs 4 and 5). The very large delay between the initial studies and the present report is surprising, but it may be understood if the present deeper analysis challenges the initial interpretation. It does not.
Anyway, the availability of more precise estimates related to previously published data does not hurt, but the manuscript as presented suffers from several misleading statements and approximations that should be corrected before publication.
a) One problem with the manuscript is the claim that “no adverse effects on renal tubular function were observed in subjects…”. Going back to ref. 5, it looks like the sub-population > 70 yrs of age does display adverse effects with significantly increased markers and decreased glomerular filtration rate, whatever the correspondence between the present area A and previously considered locations. Since the studied populations are lifelong exposed, the elderly should be the major interest of such research, with younger sub-populations being used for comparisons and for signs of onset of damage.
b) To set up the study in the Introduction, a simple map of the areas with relevant typographical features (location of previous mines and facilities, rivers, relief, etc.) and location of samples would be useful. It was likely shown before in a previous publication, but the correspondence between the 2 areas of the present work and the numerous ones from previous articles is not obvious.
c) The estimates of cadmium intake by the studied populations appear reasonable. However, the following questions pop up upon reading:
- What do the authors mean by the sentence l.155-156 (…’their Cd concentrations … by DHQ’)? How were these measurements used?
- The authors should explain why GM was used for rice and AM for other foodstuff. It seems the cadmium distribution is not the same, but it is not clear why rice is exceptional.
- l.193: is not it surprising that brewing does not extract Cd from tea leaves? Is there any explanation?
d) Area B is supposed to be more polluted (l.50-52), but the evidence is not presented or recalled, and rice GM is smaller than in area A (Table 1-a). There seems to be some discrepancy here that needs to be elaborated. The authors propose an explanation l. 290-295: “the actual accumulated levels of Cd in subjects were higher in area B than in area A, as demonstrated by their blood and urinary Cd levels (for example, urinary Cd levels in 70–79-year-old subjects were 9.34 292 and 4.90 μg/g cr. in areas A and B, respectively, and 2.99 μg/g cr. in control subjects) [5]. This discrepancy may have been due to the timing of the initiation of measures in rice farming to lower Cd absorption from the soil in 2002, namely, the flooding of paddy fields before and after heading during August”. However, this is contradicted l.289 by “The intake of Cd was higher in area A than in area B, which was attributed to differences in rice Cd concentrations”. All this is confusing, even more so l. 297: “Cd intake levels may have been higher in area B than in area A before the start of the flooding of paddy fields”. If the rice cadmium content is biased by the time of harvest, it may indeed translate into blurred blood values, but certainly not in urine ones for populations that were lifelong exposed.
e) Finally, the authors use their adjusted data to reiterate their claim that the TWI of 7 µg/kg/wk is safe. But a lot of articles reached more cautious conclusions, e.g. Urinary Cadmium Threshold to Prevent Kidney Disease Development Toxics2018, 6(2), 26, and the present day consensus tends to set a lower limit of the urinary cadmium level (5-7 µg/g creatinine) than that used by Horiguchi et al. (10 µg/g creatinine). Furthermore, some useful measurements to probe kidney function and diagnose cadmium nephropathy (aminoaciduria, glucosuria, etc.) do not seem to be available for the studied subjects. Thus, the presented data are unlikely to provide a clear-cut answer to the question, but previously published data show a clear decrease of the renal function for the most exposed women in the last part of their lives. Therefore, care must prevail in stating conclusions. Indeed the TWI is today far lower than the Japanese value in other parts of the world, and it is provocative to write that “ The present results validated the previous PTWI of CODEX or the current TWI of Japan of 7 μg/kg BW/week as a safety standard to protect human health against Cd”. Such statements can even been dangerous as a signal to regulatory bodies, particularly concerning women in Japan who have a very high life expectancy.
These statements have to be changed in the present manuscript, including in the Abstract. The authors should fully revise their manuscript keeping in mind the limitations of their work and strongly avoiding overstating what they can conclude from it.
Minor
l.38 convert ‘renal osteomalacia’ to ‘osteomalacia’ only
avoid using plural for intake e.g. l.124-126; 217, and elsewhere
l.219: correct ‘by whom’ to ‘for whom’
Author Response
Responses to Reviewer 3’s Comments
Thank you very much for your valuable comments, according to which we revised the manuscript. Point-by-point responses are provided below. The parts that we changed are highlighted in the manuscript.
Point 1: In this manuscript, the authors reconsider food samples obtained in the early ‘00s to estimate the exposure to cadmium of populations that were studied then, with results published long ago (refs 4 and 5). The very large delay between the initial studies and the present report is surprising, but it may be understood if the present deeper analysis challenges the initial interpretation. It does not.
Response 1: The purpose of the present study was not to show the current situation in Cd-polluted areas in Akita, but to examine previous exposure to high Cd levels in local farmers. We did not submit the data collected at that time for publication. Nevertheless, the results obtained remain important, even after approximately 20 years, because few Cd-polluted areas and subjects exposed to high levels of Cd remain in Japan. Furthermore, data were collected at the start of the flooding of paddy fields, which has successfully decreased Cd exposure levels in farmers in these areas; therefore, these data will never be obtained again in the future.
We added the following information to the Discussion, “Since the data used were obtained in 2001–2003, the current situation in these Cd-polluted areas remains unclear. Nevertheless, the results obtained remain important, even after approximately 20 years, because few Cd-polluted areas remain in Japan. Furthermore, data were collected at the start of the flooding of paddy fields, which has successfully decreased Cd exposure levels in farmers in these areas; therefore, these data will never be obtained again in the future.” on lines 410–415.
Point 2: One problem with the manuscript is the claim that “no adverse effects on renal tubular function were observed in subjects…”. Going back to ref. 5, it looks like the sub-population > 70 yrs of age does display adverse effects with significantly increased markers and decreased glomerular filtration rate, whatever the correspondence between the present area A and previously considered locations. Since the studied populations are lifelong exposed, the elderly should be the major interest of such research, with younger sub-populations being used for comparisons and for signs of onset of damage.
Response 2: We added age-classified weekly cadmium intake per body weight as Table 5 and “We further divided weekly Cd intake per body weight into age-classified groups and examined differences between them (Table 5). In area A, weekly Cd intake levels per body weight were higher in subjects aged 40 or older than in younger subjects, while no significant differences were observed between age-classified groups in area B. Weekly Cd intake levels per body weight in subjects aged 50 or older were above PTWI or TWI in area A.” on lines 285–289. Consequently, “Table 5” was changed to “Table 6”.
We also added “Weekly Cd intake per body weight was above the PTWI or TWI in older subjects in area A; however, there were no age-classified subgroups above the PTWI or TWI in area B, which may have been due to a decrease in Cd concentrations in rice, suggesting a higher risk of developing renal tubular dysfunction among the elderly.” on lines 370–373, and “Regarding multiple comparisons, the Steel–Dwass test was performed to compare the medians of age-classified Cd intake.” to Statistical analysis on lines 188–189.
Point 3: To set up the study in the Introduction, a simple map of the areas with relevant typographical features (location of previous mines and facilities, rivers, relief, etc.) and location of samples would be useful. It was likely shown before in a previous publication, but the correspondence between the 2 areas of the present work and the numerous ones from previous articles is not obvious.
Response 3: We added a map and its accompanying legend as Figure 1. Consequently, the numbers of figures changed.
Point 4: What do the authors mean by the sentence l.155-156 (…’their Cd concentrations … by DHQ’)? How were these measurements used?
Response 4: We changed “Cd intakes from garlic, okra, belvedere fruit, kelp, hijiki seaweed, agar-agar, mozuku seaweed, and scallops with innards were also excluded; however, their Cd concentrations were measured because their intakes were not evaluated by DHQ.” to “Cd intake from garlic, okra, belvedere fruit, kelp, hijiki seaweed, agar-agar, mozuku seaweed, and scallops with innards was also excluded because they were not evaluated by DHQ; however, their Cd concentrations were measured.” on lines 166–169.
Point 5: The authors should explain why GM was used for rice and AM for other foodstuff. It seems the cadmium distribution is not the same, but it is not clear why rice is exceptional.
Response 5 Since Cd concentrations follow a clear lognormal distribution because of large numbers, and the distribution of Cd concentrations in other foodstuffs was not clear because of small numbers, we considered it appropriate to use GM for rice and AM for other foodstuffs to calculate Cd intake.
Therefore, we changed “Cd concentrations in rice were presented as GMs with 25th and 75th percentiles.” to “Since Cd concentrations in rice followed a clear lognormal distribution with large numbers and the distribution of Cd concentrations in other food items was not clear because of small numbers, rice and other food items were shown as GMs and AMs, respectively.” on lines 182–185.
Point 6: l.193: is not it surprising that brewing does not extract Cd from tea leaves? Is there any explanation?
Response 6: Brewing DOES extract Cd from tea leaves; therefore, we did not use it for the calculation of Cd intake (line 166).
We added “(data not shown)” on line 210, changed “and those from livestock food and fruit were included as the subgroup of others” to “and that from in the subgroup as others, while that from fruit was excluded from calculations” on lines 242–244, and changed “including livestock food, fruit, chocolate, and flavor seasonings, and tea leaves” to “including manju, livestock food, chocolate, and flavor seasonings.” in Table 3.
Point 7: Area B is supposed to be more polluted (l.50-52), but the evidence is not presented or recalled, and rice GM is smaller than in area A (Table 1-a). There seems to be some discrepancy here that needs to be elaborated. The authors propose an explanation l. 290-295: “the actual accumulated levels of Cd in subjects were higher in area B than in area A, as demonstrated by their blood and urinary Cd levels (for example, urinary Cd levels in 70–79-year-old subjects were 9.34 292 and 4.90 μg/g cr. in areas A and B, respectively, and 2.99 μg/g cr. in control subjects) [5]. This discrepancy may have been due to the timing of the initiation of measures in rice farming to lower Cd absorption from the soil in 2002, namely, the flooding of paddy fields before and after heading during August”. However, this is contradicted l.289 by “The intake of Cd was higher in area A than in area B, which was attributed to differences in rice Cd concentrations”. All this is confusing, even more so l. 297: “Cd intake levels may have been higher in area B than in area A before the start of the flooding of paddy fields”. If the rice cadmium content is biased by the time of harvest, it may indeed translate into blurred blood values, but certainly not in urine ones for populations that were lifelong exposed.
Response 7: The biological half-life of Cd in humans is very long, as much as 10–30 years. Urinary Cd levels generally reflect life-long accumulation in the kidneys, while blood Cd levels indicate recent Cd exposure. However, in the case of high Cd exposure, blood and urinary Cd levels are both useful indicators of the accumulation of Cd in the body. Therefore, we added “Since the biological half-life of Cd in humans is very long (10–30 years), blood and urinary Cd levels are stable indicators of the accumulation of Cd in the human body, particularly when the level of Cd that accumulates is high [1].” on lines 322–325.
Point 8: Finally, the authors use their adjusted data to reiterate their claim that the TWI of 7 µg/kg/wk is safe. But a lot of articles reached more cautious conclusions, e.g. Urinary Cadmium Threshold to Prevent Kidney Disease Development Toxics2018, 6(2), 26, and the present day consensus tends to set a lower limit of the urinary cadmium level (5-7 µg/g creatinine) than that used by Horiguchi et al. (10 µg/g creatinine). Furthermore, some useful measurements to probe kidney function and diagnose cadmium nephropathy (aminoaciduria, glucosuria, etc.) do not seem to be available for the studied subjects. Thus, the presented data are unlikely to provide a clear-cut answer to the question, but previously published data show a clear decrease of the renal function for the most exposed women in the last part of their lives. Therefore, care must prevail in stating conclusions. Indeed the TWI is today far lower than the Japanese value in other parts of the world, and it is provocative to write that “ The present results validated the previous PTWI of CODEX or the current TWI of Japan of 7 μg/kg BW/week as a safety standard to protect human health against Cd”. Such statements can even been dangerous as a signal to regulatory bodies, particularly concerning women in Japan who have a very high life expectancy.
Response 8: The main purpose of the present study was to verify oral Cd exposure levels in farmers who received health examinations in our previous study, and not to conduct a general risk assessment of Cd.
Therefore, we changed “Since there were no affected subjects in area A, the present results validated the PTWI or TWI as a safety standard of cadmium.” to “These results demonstrated that farmers exposed to cadmium were at risk of adverse effects.” in the Abstract, changed “which was consistent with our previous findings, thereby demonstrating their validity as a safety standard” to “which showed that subjects exposed to Cd were at risk of adverse effects.” on line 78, changed “These results indicate that previously estimated Cd intake levels, based only on Cd concentrations in rice and miso, were accurate. Since no adverse effects on renal tubular function were observed in subjects in area A, the appropriateness of the PTWI of CODEX at that time or the current TWI in Japan of 7 µg/kg BW/week as a safety standard to protect human health against Cd has been reconfirmed.” to “These results indicate that previously estimated Cd intake levels, based only on Cd concentrations in rice and miso, were not inaccurate, and that local farmers in Cd-polluted areas in Akita exposed to Cd were at risk of adverse effects.” on lines 367–370, and changed “…that their Cd intake, which was higher than that by the general population, were derived from local agricultural products, particularly rice. The present results validated the previous PTWI of JECFA or the current TWI of Japan of 7 µg/kg BW/week as a safety standard to protect human health against Cd.” to “that Cd intake was higher than those by the general population, which were derived from local agricultural products, particularly rice, and that their exposure to Cd increased the risk of adverse effects.” in the Conclusion on lines 432–435.
Point 9: l.38 convert ‘renal osteomalacia’ to ‘osteomalacia’ only
Response 9: We changed “renal osteomalacia” to “osteomalacia” on line 39.
Point 10: avoid using plural for intake e.g. l.124-126; 217, and elsewhere
Response 10: We have avoided using “intakes”.
Point 11: l.219: correct ‘by whom’ to ‘for whom’
Response 11: We changed “by whom” to “for whom” on line 236.
# Other corrected parts:
We changed “Cadmium-polluted areas exist in Akita prefecture, located in the northern part of Japan.” to “Akita prefecture, located in the northern part of Japan, has many cadmium-polluted areas.”on lines 19–20.
We changed “We performed an exposure assessment of cadmium on 712 and 432 female farmers” to “We herein performed an exposure assessment of cadmium in 712 and 432 female farmers” on line 20.
We changed “of” to “reported by” on line 28.
We changed “of Japan” to “in Japan” on line 29.
We changed “there are still large, but scattered, Cd-polluted areas in Akita prefecture, located in the northern part of Japan, due to the previous activities of mines and smelters” to “large, but scattered, Cd-polluted areas still remain in Akita prefecture, which is located in the northern part of Japan, due to the previous activities of mines and smelters” on lines 45–47.
We changed “and” to “while” on line 53.
We changed “of Cd on subjects on Cd in subjects” to “” on line 62.
We changed “the inhabitants” to “its inhabitants” on line 68.
We changed “verify” to “confirm” on line 77.
We changed “Labor” to “Labour” on line 89.
We changed “in the diet surveys” to “in diet surveys” on line 90.
We changed “the Japan Food Research Laboratories” to “Japan Food Research Laboratories” on lines 104–105.
We changed “In total, 0.4 and 0.8 µg/mL standard Cd solutions were made by diluting the original standard Cd solution (Kanto Chemical Co., Inc., Tokyo, Japan) with 1% hydrochloric acid (HCl).” to “” on lines 118–120.
We changed “food” to “food items” on lines 112–113.
We changed “and” to “while” on line 149.
We changed “row” to “raw” on line 156.
We changed “The intakes of mushrooms were” to “The intake of mushrooms was” on line 157.
We changed “cod” to “of cod” on line 160.
We changed “by DHQ” to “in DHQ on line 161.
We changed “Cd concentrations” to “the Cd concentrations” on line 161.
We changed “scallop” to “scallops” on line 162.
We changed “to their corresponding food intakes” to “by their corresponding food intake” on line 170.
We changed “of individual subjects” to “in individual subjects” on lines 170– 171.
We changed “Cd concentrations” to “the Cd concentration” on line 175.
We changed “the standard tables of food compositions” to “the Standard Tables of Food Compositions” on line 177.
We changed “that their distributions were log-normal” to “of a lognormal distribution” on line 191.
We changed “These statistical analyses” to “Statistical analyses” on line 192.
We changed “Lists of food items and their Cd concentrations, divided into 10 subgroups, are shown in Table 1.” to “Food items and their Cd concentrations, divided into 10 subgroups, are listed in Table 1.” on line 195.
We changed “less” to “lower” on line 197.
We changed “, but 8.2% and 5.8% of Cd concentrations in rice were over the safety standard in areas A and B, respectively” to “however, Cd concentrations of 8.2 and 5.8% in rice in areas A and B, respectively, were above the safety standard” on lines 197–198.
We changed “is” to “are” on line 212.
We changed “Dropworta” to “Dropwort” in table 1-d.
We changed “. There were statistically significant differences” to “, whereas significant differences were noted” on lines 238–239.
We changed “between them, but the differences are very small and biologically negligible” to “, but were biologically negligible.” on line 239.
We changed “were” to “was” on line 242.
We changed “in” to “from” on line 244.
We changed “total Cd intake” to “he total Cd intake” on lines 245–246.
We changed “in” to “from” on line 246.
We changed “7.2 µg/kg BW/week in area A and 6.0 µg/kg BW/week in area B, both of which were approximately PTWI or TWI” to “7.2 and 6.0 µg/kg BW/week in areas A and B, respectively, with both being approximately PTWI or TWI” on lines 272–273.
We changed “distribution” to “distributions” on line 273.
We changed “If data obtained from 2 subjects in area A that were extremely large and considered to be outliers are excluded, the distributions in both areas became similar and skewed to the higher side.” to “The exclusion of data obtained from 2 subjects in area A that were extremely large and considered to be outliers resulted in similar distributions in both areas that skewed to the higher side.” on lines 274–276.
We changed “Cd intakes, 7.0 and 6.0” to “Cd intake levels of 7.0 and 6.0 µg/kg BW/week” on lines 294–295.
We changed “very close” to “similar” on line 295.
We changed “Cd concentrations” to “their Cd concentrations” on line 306.
We changed “high levels of Cd” to “high Cd levels” on line 307.
We changed “The results obtained” to “The present results” on line 308.
We changed “Cd intakes were” to “Cd intake was” on line 309.
We changed “by our subjects than by the general population” to “in our subjects than in the general population” on line 309.
We changed “weekly Cd intakes per body weight, which were verified by the Monte Carlo simulation, were approximately the same levels of the PTWI of JECFA or TWI of Japan, and 38.0–51.7% of the subjects’ weekly Cd intakes per body weight were over it” to “weekly Cd intake per body weight, which was confirmed by the Monte Carlo simulation, were approximately the same as the PTWI of JECFA or TWI of Japan, while 38.0–51.7% of subjects had weekly Cd intake per body weight that was above it” on lines 310–312.
We changed “there were rice with Cd concentrations over the safety standard” to “rice also had Cd concentrations that were above the safety standard” on lines 317–318.
We changed “were” to “was” on line 334.
We changed “were” to “was” on line 335.
We changed “were” to “was” on line 337.
We changed “in general Japanese” to “the general Japanese population” on line 338.
We changed “were” to “was” on line 338.
We changed “in general Japanese” to “the general Japanese population” on line 342.
We changed “Cd concentrations in sesame seeds, taro (satoimo in Japanese), yams (yamaimo in Japanese), soybeans, carrot, spinach, burdock (gobo in Japanese), garland chrysanthemum (shungiku in Japanese), Japanese mustard spinach (komatsuna in Japanese), garlic, belvedere fruit (tomburi in Japanese), and shiitake mushroom were high” to “Cd concentrations were high in sesame seeds, taro (satoimo in Japanese), yams (yamaimo in Japanese), soybeans, carrot, spinach, burdock (gobo in Japanese), garland chrysanthemum (shungiku in Japanese), Japanese mustard spinach (komatsuna in Japanese), garlic, belvedere fruit (tomburi in Japanese), and shiitake mushroom” on lines 349–352.
We changed “Cd concentrations in livestock food and fruit were generally low” to “Cd concentrations were generally low in livestock food and fruit” on line 356.
We changed “Cd concentrations in tubers, soybeans, brightly colored vegetables, and seafood, particularly seaweed, and the innards of squid or shellfish were high in Japan” to “Cd concentrations were high in tubers, soybeans, brightly colored vegetables, and seafood, particularly seaweed, and the innards of squid or shellfish in Japan” on lines 357–359.
We changed “The median weekly Cd intakes” to “Median weekly Cd intake levels” on line 360.
We changed “the chemical concentrations in individual foodstuffs remain unknown” to “the concentration of this chemical in individual food items remains unknown” on lines 378–379.
We changed “of individuals” to “by individuals” on line 380.
We changed “the present study” to “the present results” on line 382.
We changed “In general, the individual food analysis” to “The individual food analysis generally” on line 386.
We changed “In addition” to “Furthermore” on line 388.
We changed “the standard tables of food compositions” to “the Standard Tables of Food Compositions” on lines 389–390.
We changed “The subjects” to “Furthermore, the subjects” on line 416.
We changed “remain” to “remains” on line 417.
We changed “do not reflect” to “did not reflect” on line 421.
We changed “on female farmers” to “in female farmers” on lines 431.

Round 2
Reviewer 1 Report
I am not convinced of the author’s response (Response 1). I still think the same. The study is very old and the data is 20 years old.
Even though (according with authors) these data were worth being published (“because few Cd-polluted areas and subjects exposed to high levels of Cd remain in Japan”) this would only have sense in case that these data had been compared with current data. I mean, repeat the study in the same areas 20 year later and compare the obtained results. I suggest the authors do it
My feeling is that authors should have published these data many years ago and now they want to published them in an open - access journal notwithstanding.
The paper must be rejected
Author Response
>Even though (according with authors) these data were worth being published (“because few Cd-polluted areas and subjects exposed to high levels of Cd remain in Japan”) this would only have sense in case that these data had been compared with current data. I mean, repeat the study in the same areas 20 year later and compare the obtained results. I suggest the authors do it
The individual food analysis method used in this study, purchasing every local foodstuff and measuring Cd concentrations in all of them, required lots of time, manpower, and funds. It is impossible to do the same study today in the areas again.
However, since rice is a main contributor for total Cd intake in Japanese, it is reasonable to assume recent Cd intake levels among farmers in the areas based on recent Cd concentrations in their locally produced and consumed rice. Actually, as demonstrated in this paper, the Cd intake levels assumed from only rice and miso in the previous study were similar to those of the current study.
Therefore, we presented annual trend of rates of rice with a cadmium concentration above the safety standard around the start of the flooding of paddy fields as a supplemental table, and our own data of cadmium concentrations in rice after that.
As a results, we changed “Actually, the rates of rice with Cd concentration above the safety standard in these areas were decreased after that; 0% (n=8), 3.3% (n=183), and 0% (n=10) in Odate, Kazuno, and Kosaka, respectively, in 2008, according to the results of investigations targeted for Cd-polluted rice fields by Akita prefecture (Akita Prefect. Dep. of Agriculture, Forestry and Fisheries).” to “Actually, the rates of rice with Cd concentration above the safety standard in these areas were decreased after that according to the results of intensive inspections targeted for Cd-polluted rice fields by Akita prefecture (Akita Prefect. Dep. of Agriculture, Forestry and Fisheries) (supplemental table).” on line 343–346, and added “More recently, our own investigations on farmers in the areas show that the median of Cd concentration in their self-harvested rice from 2010 to 2018 was 0.096 mg/kg and 0.7% of them were above the safety standard (n=599) (unpublished data). These results indicate the flooding of paddy fields have effectively lowered Cd concentrations in rice from 2002 in these areas.” on line 346–350.
In addition, we changed “Since the data used were obtained in 2001–2003, the current situation in these Cd-polluted areas remains unclear.” to “The data used in this study were obtained in 2001–2003. ” on line 434–435.
Reviewer 3 Report
The authors have considered the criticism on the submitted version and they have significantly improved this manuscript. Yet, some points remain obscure and would need to be clarified before acceptance.
Last sentence of abstract: “These results demonstrated that farmers exposed to cadmium were at risk of adverse effects.” The present data actually demonstrate nothing: they are simply confirming the risk assessment of a previous publication with more detailed estimate of the cadmium intake. Similarly, in the last part of the concluding sentence l.428: the data in the present manuscript do not “show … that their exposure to Cd increased the risk of adverse effects” since the health data were published before and have not been included in the present analysis. It would be better to leave this aside.
l.193-194 and 311-313: what is the meaning of the added sentence in the Result section? Why presenting concentrations in mg/kg in a sentence and % in the next? The sentences in the Discussion do not clarify the issue.
If brewing does extract Cd from tea leaves (cover letter), why neglecting it in calculating the intake (manuscript) of supposedly heavy tea drinkers? The sentence l.205-206 is thus obscure.
Figures 3 and 4 show inverted patterns in intensities: what is the origin of this difference between the experimental estimates and the probabilities in simulations? Are the 2 outliers in A (in more than 700 subjects!) enough to justify the difference?
l.366-367: the suggested link between age-stratified analysis and health risk in area B cannot be readily understood: please clarify or remove.
Other issues:
Abstract l.20 add a comma after …female farmers
l.39: Renal anemia makes no sense. Anemia is the condition.
l.45: change ‘still remain’ by ‘exist’ unless some remediation action was also taken in the Akita prefecture as in the Toyama one.
Figure 1 has been added: it should be called in the text, preferably l.50-53.
l.69: “we newly collected ..” That is fairly confusing since the collection dates back to the early 2000s.
l.183: proportion should be plural.
l.360: remove ‘of’
l.397: countries
l.421-423: clarify this sentence by: “Although median values in Cd-polluted areas cannot be statistically compared to these averages, the differences are large enough to identify areas at risk of health problems.” for instance.
Author Response
>Last sentence of abstract: “These results demonstrated that farmers exposed to cadmium were at risk of adverse effects.” The present data actually demonstrate nothing: they are simply confirming the risk assessment of a previous publication with more detailed estimate of the cadmium intake. Similarly, in the last part of the concluding sentence l.428: the data in the present manuscript do not “show … that their exposure to Cd increased the risk of adverse effects” since the health data were published before and have not been included in the present analysis. It would be better to leave this aside.
We changed “These results demonstrated that farmers exposed to cadmium were at risk of adverse effects.” to “These results demonstrated that the cadmium exposure levels among the farmers were high enough to be approximately the tolerable weekly intake.” on lines 32–33 in Abstract.
And, we changed “that their exposure to Cd increased the risk of adverse effects” to “that their exposure levels to Cd were approximately the PTWI of JECFA or TWI of Japan” on lines 458–459.
>l.193-194 and 311-313: what is the meaning of the added sentence in the Result section? Why presenting concentrations in mg/kg in a sentence and % in the next? The sentences in the Discussion do not clarify the issue.
We changed “Cd concentrations of 8.2 and 5.8% in rice in areas A and B, respectively, were above the safety standard” to “8.2 and 5.8% of rice had Cd concentrations that were above the safety standard in areas A and B, respectively.” on lines 211–213.
And, we changed “rice also had Cd concentrations that were above the safety standard” to “there were rice that had Cd concentrations above the safety standard” on lines 333–334.
>If brewing does extract Cd from tea leaves (cover letter), why neglecting it in calculating the intake (manuscript) of supposedly heavy tea drinkers? The sentence l.205-206 is thus obscure.
We are sorry, brewing does not extract Cd from tea leaves. Cd was not detected in hot water after brewing tea leaves, and probably most Cd remained in the tea leaves.
We excluded “(data not shown)” on line 225, which is unnecessary.
We changed “Japanese green tea” to “Japanese green tea leaves” in Table 1-j for clarification.
>Figures 3 and 4 show inverted patterns in intensities: what is the origin of this difference between the experimental estimates and the probabilities in simulations? Are the 2 outliers in A (in more than 700 subjects!) enough to justify the difference?
Figures 3 and 4 do not show inverted patterns in intensities. Although the peak of probability in area B is higher than that in area A, careful inspection would make you see that the value (weekly Cd intake) of the peak in area B is smaller than that in area A, and probability is distributed to a smaller value side more in area B than in area A.
>l.366-367: the suggested link between age-stratified analysis and health risk in area B cannot be readily understood: please clarify or remove.
We changed “Weekly Cd intake per body weight was above the PTWI or TWI in older subjects in area A; however, there were no age-classified subgroups above the PTWI or TWI in area B, which may have been due to a decrease in Cd concentrations in rice, suggesting a higher risk of developing renal tubular dysfunction among the elderly.” to “Weekly Cd intake per body weight was above the PTWI or TWI in older subjects in area A, suggesting a higher risk of developing renal tubular dysfunction among the elderly. There were no age-classified subgroups above the PTWI or TWI in area B, but actually the older subjects might had been exposed to much higher levels of Cd like in area A before the start of the flooding of paddy fields.” on lines 393–397.
Other issues:
>Abstract l.20 add a comma after …female farmers
We added a comma after that.
>l.39: Renal anemia makes no sense. Anemia is the condition.
We are afraid that almost all anemia observed in patients with itai-itai disease is renal anemia. Renal anemia is derived from insufficient production of erythropoietin in the kidney, and the erythropoietin producing cells in the kidney are targeted by Cd. That is why renal anemia is an intrinsic clinical character in itai-itai disease along with osteomalacia.
>l.45: change ‘still remain’ by ‘exist’ unless some remediation action was also taken in the Akita prefecture as in the Toyama one.
In Akita, some restorations of Cd-polluted paddy fields have been done if the pollution levels were severe.
>Figure 1 has been added: it should be called in the text, preferably l.50-53.
We added “(Figure 1)” on line 54.
>l.69: “we newly collected ..” That is fairly confusing since the collection dates back to the early 2000s.
We deleted “newly” on line 73.
>l.183: proportion should be plural.
We changed “proportion” to “proportions” on line 201.
>l.360: remove ‘of’
We removed ‘of’ on line 389.
>l.397: countries
We cannot understand what to do about this “countries”.
>l.421-423: clarify this sentence by: “Although median values in Cd-polluted areas cannot be statistically compared to these averages, the differences are large enough to identify areas at risk of health problems.” for instance.
We changed “Although it may be statistically inappropriate to compare averages in the general population with medians in Cd-polluted areas, it would not be a serious problem because of the large difference between them.” to “Although median values in Cd-polluted areas cannot be statistically compared to these averages, the differences are large enough to identify areas at risk of health problems.” on lines 450–452.
# Other corrected part:
We changed “Total Cd intake” to “Total cadmium intake” in Table 3.